# Structural basis for effector transmembrane domain recognition by type VI secretion system chaperones

Shehryar Ahmad[1,2], Kara K Tsang[1,2†], Kartik Sachar[3†], Dennis Quentin[4], Tahmid M Tashin[1,2], Nathan P Bullen[1,2], Stefan Raunser[4], Andrew G McArthur[1,2,5], Gerd Prehna[3*], John C Whitney[1,2,5*]

[1]Michael DeGroote Institute for Infectious Disease Research, McMaster University, Hamilton, Canada; [2]Department of Biochemistry and Biomedical Sciences, McMaster University, Hamilton, Canada; [3]Department of Microbiology, University of Manitoba, Winnipeg, Canada; [4]Department of Structural Biochemistry, Max Planck Institute of Molecular Physiology, Dortmund, Germany; [5]David Braley Centre for Antibiotic Discovery, McMaster University, Hamilton, Canada

**Abstract** Type VI secretion systems (T6SSs) deliver antibacterial effector proteins between neighboring bacteria. Many effectors harbor N-terminal *trans*membrane *do*mains (TMDs) implicated in effector translocation across target cell membranes. However, the distribution of these TMD-containing effectors remains unknown. Here, we discover prePAAR, a conserved motif found in over 6000 putative TMD-containing effectors encoded predominantly by 15 genera of Proteobacteria. Based on differing numbers of TMDs, effectors group into two distinct classes that both require a member of the Eag family of T6SS chaperones for export. Co-crystal structures of class I and class II effector TMD-chaperone complexes from *Salmonella* Typhimurium and *Pseudomonas aeruginosa*, respectively, reveals that Eag chaperones mimic transmembrane helical packing to stabilize effector TMDs. In addition to participating in the chaperone-TMD interface, we find that prePAAR residues mediate effector-VgrG spike interactions. Taken together, our findings reveal mechanisms of chaperone-mediated stabilization and secretion of two distinct families of T6SS membrane protein effectors.

**\*For correspondence:**
gerd.prehna@umanitoba.ca (GP); jwhitney@mcmaster.ca (JCW)

†These authors contributed equally to this work

**Competing interests:** The authors declare that no competing interests exist.

## Introduction

Bacteria secrete proteins to facilitate interactions with their surrounding environment. In Gram-negative bacteria, the transport of proteins across cellular membranes often requires the use of specialized secretion apparatuses found within the cell envelope. One such pathway is the type VI secretion system (T6SS), which in many bacterial species functions to deliver antibacterial effector proteins from the cytoplasm directly into an adjacent bacterial cell via a one-step secretion event (*Russell et al., 2011*). A critical step that precedes type VI secretion is the selective recruitment of effectors to the T6SS apparatus. Recent work has shown that for many effectors this process requires chaperone proteins, which are thought to maintain effectors in a 'secretion-competent' state (*Unterweger et al., 2017*). However, to-date, no molecular-level evidence exists to support this idea.

The T6SS is comprised of two main components: a cell envelope-spanning membrane complex and a cytoplasmic bacteriophage tail-like complex. The latter contains a tube structure formed by many stacked copies of hexameric ring-shaped *h*emolysin *co*-regulated *p*rotein (Hcp) capped by a single homotrimer of *v*aline-*g*lycine *r*epeat protein *G* (VgrG) (*Mougous et al., 2006*; *Spínola-Amilibia et al., 2016*). Together, these proteins form an assembly that resembles the tail-tube and

spike components of contractile bacteriophage (*Renault et al., 2018*). Additionally, VgrG proteins interact with a single copy of a cone-shaped *p*roline-*a*lanine-*a*lanine-*a*rginine (PAAR) domain-containing protein that forms the tip of the VgrG spike (*Shneider et al., 2013*). Altogether, PAAR, Hcp and VgrG are necessary for T6SS function, and during a secretion event these components are themselves delivered into target cells (*Cianfanelli et al., 2016a*). Prior to its export from the cell, the bacteriophage tail-like complex is loaded with toxic effector proteins. In contrast to proteins that are exported by the general secretory pathway, T6SS effectors do not contain linear signal sequences that facilitate their recognition by the T6SS apparatus. Instead, effectors transit the T6SS via physical association with Hcp, VgrG, or PAAR proteins (*Cianfanelli et al., 2016b*).

In addition to its role in effector export, Hcp also possesses chaperone-like properties that facilitate cytoplasmic accumulation of Hcp-interacting effectors prior to their secretion (*Silverman et al., 2013*). This chaperone activity has been attributed to the interior of the ~4 nm pore formed by hexameric Hcp rings, which are wide enough to accommodate small, single-domain effectors. Individual Hcp rings appear to possess affinity toward multiple unrelated effectors. However, the molecular basis for this promiscuous substrate recognition is unknown.

In contrast to their Hcp-associated counterparts, VgrG-linked effectors are typically comprised of multiple domains and often require effector-specific chaperones for stability and/or to facilitate their interaction with the VgrG spike. Thus far, three effector-specific chaperone families belonging to the DUF1795, DUF2169, and DUF4123 protein families have been described. Studies on representative DUF2169 and DUF4123 proteins indicate that these chaperones minimally form ternary complexes with their cognate effector and a PAAR protein to facilitate the 'loading' of the PAAR domain and effector onto their cognate VgrG (*Bondage et al., 2016*; *Burkinshaw et al., 2018*). In contrast, DUF1795 proteins, also known as *e*ffector-*a*ssociated *g*ene (Eag) chaperones, interact with so-called 'evolved' PAAR proteins in which the PAAR and toxin domains are found as a single polypeptide chain (*Whitney et al., 2015*; *Alcoforado Diniz and Coulthurst, 2015*). Biochemical characterization of the Eag chaperone EagT6 from *P. aeruginosa* found that this chaperone interacts with TMDs found in the N-terminal loading and translocation region of its associated effector, Tse6 (*Quentin et al., 2018*). In the presence of lipid vesicles, Tse6 spontaneously inserts into membranes causing EagT6 chaperones to be released suggesting that EagT6 maintains the N-terminal TMDs in a pre-insertion state prior to toxin domain delivery across the inner membrane of target bacteria. However, it is not known whether the 'solubilization' of TMDs in aqueous environments represents a general role for Eag chaperones and if so, it is unclear how they maintain effector TMDs in a pre-insertion state.

In this work, we report the identification of prePAAR, a highly conserved motif that enabled the identification of over 6000 putative T6SS effectors, all of which possess N-terminal TMDs and co-occur in genomes with Eag chaperones. Further informatics analyses found that these candidate effectors can be categorized into one of two broadly defined classes. Class I effectors belong to the Rhs family of proteins, are comprised of ~1200 amino acids, and possess a single region of N-terminal TMDs. Class II effectors are ~450 amino acids in length and possess two regions of N-terminal TMDs. We validate our informatics approach by showing that a representative member of each effector class requires a cognate Eag chaperone for T6SS-dependent delivery into susceptible bacteria. Crystal structures of Eag chaperones in complex with the TMDs of cognate class I and class II effectors reveal the conformation of effector TMDs prior to their secretion and insertion into target cell membranes. In addition to participating in chaperone-effector interactions, structure-guided mutagenesis of hydrophilic residues within prePAAR show that this motif also enables effector interaction with its cognate VgrG. Collectively, our data provide the first high-resolution structural snapshots of T6SS effector-chaperone interactions and define the molecular determinants for effector TMD stabilization and recruitment to the T6SS apparatus.

## Results

### prePAAR is a motif found in TMD-containing effectors that interact with Eag chaperones

Characterization of Eag chaperones and their associated effectors has thus far been limited to the EagT6-Tse6 and EagR1-RhsA chaperone-effector pairs from *P. aeruginosa* and *Serratia marcescens*,

respectively (*Cianfanelli et al., 2016a*; *Whitney et al., 2015*). In both cases, the chaperone gene is found upstream of genes encoding its cognate effector and an immunity protein that protects the toxin-producing bacterium from self-intoxication (*Figure 1A*). We previously showed that EagT6 interacts with the N-terminal TMDs of Tse6, an observation that led us to hypothesize a general role for Eag chaperones in 'solubilizing' hydrophobic TMDs of effectors in the aqueous environment of the cytoplasm so they can be loaded into the T6SS apparatus (*Figure 1B*; *Quentin et al., 2018*). However, evidence supporting this general role is lacking because homology-based searches for

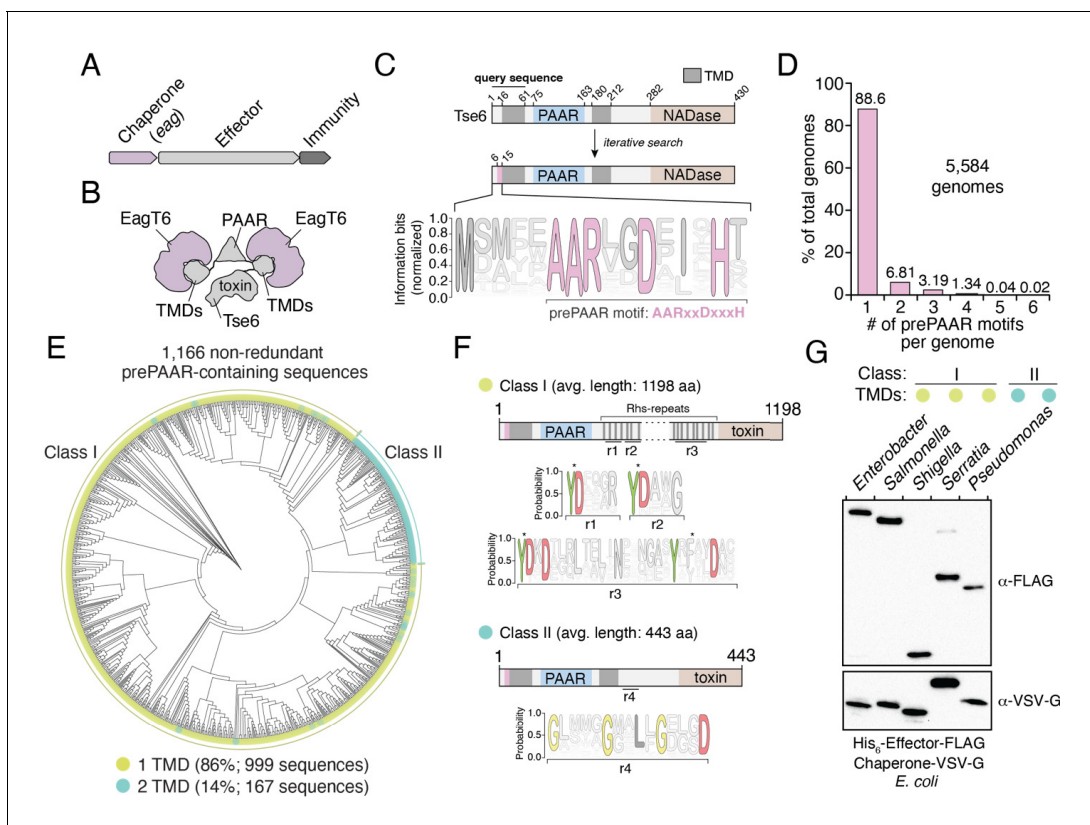

**Figure 1.** The prePAAR motif is conserved across multiple bacterial genera and is found in T6SS effectors that interact with Eag chaperones. (**A**) Genomic arrangement of T6SS chaperone-effector-immunity genes for characterized *e*ffector-*a*ssociated *g*ene family members (*eag*; shown in purple), which encode DUF1795 domain-containing chaperones. (**B**) Schematic depicting Eag chaperone interactions with the transmembrane domain (TMD) regions of the model chaperone-effector pair EagT6-Tse6. (**C**) Protein architecture and sequence logo for the prePAAR motif found in the N-terminus of Tse6. An alignment of 2,054 sequences was generated using the 61 N-terminal residues of Tse6 as the search query. The relative frequency of each residue and information content in bits was calculated at every position of the sequence and then normalized by the sum of each position's information bits. Transparency is used to indicate probability of a residue appearing at a specific position. Residues colored in pink correspond to the prePAAR motif: AARxxDxxxH. (**D**) Genomes from genera of Proteobacteria known to contain functional T6SSs (*Burkholderia*, *Escherichia*, *Enterobacter*, *Pseudomonas*, *Salmonella*, *Serratia*, *Shigella*, *Yersinia*) were screened for unique prePAAR effectors. Percentage of total genomes that contained 1 to 6 prePAAR motifs is indicated. (**E**) Phylogenetic distribution of 1,166 non-redundant prePAAR-containing effectors identified in **B**. TM prediction algorithms were used to quantify the number of TM regions in each effector. The two classes that emerged are labeled in green (class I; 1 TM region-containing effectors) and blue (class II; 2 TM region-containing effectors). Branch lengths indicates evolutionary distances. (**F**) Effector sequences within class I or class II were aligned, and a sequence logo was generated based on the relative frequency of each residue at each position to identify characteristic motifs of both classes. Four different regions (r1–r4) after the PAAR and TM regions were found to harbor conserved residues. Class I effectors contain YD repeat regions (r1-3) characteristic of Rhs proteins, whereas a GxxxxGxxLxGxxxD motif (r4) was identified in class II effectors. (**G**) Western blot analysis of five effector-chaperone pairs that were selected from the indicated genera, based on the analysis in **B**. Each pair was co-expressed in *E. coli* and co-purified using nickel affinity chromatography. The class and number of TM regions from each pair are indicated. Locus tags for each pair (e, effector; c, chaperone) are as follows: *Enterobacter* (e: ECL_01567, c: ECL_01566), *Shigella* (e: SF0266, c: SF3490), *Salmonella* (e: SL1344_0286, c: SL1344_0285), *Serratia* (e: Spro_3017, c: Spro_3016), *Pseudomonas* (e: PA0093, c: PA0094). Note that the Rhs component of the class I prePAAR effector SF0266 is encoded by the downstream open-reading frame SF0267 (see Figure 1—figure supplement 1C for details).

The online version of this article includes the following figure supplement(s) for figure 1:

**Figure supplement 1.** prePAAR effectors contain a fixed number of transmembrane domains.

additional Eag chaperones can yield difficult to interpret results due to a scarcity of conserved residues and homology of this protein family to the phage protein DcrB (*Samsonov et al., 2002*), which is widely distributed in both T6SS-positive and T6SS-negative organisms. Similarly, the identification of N-terminal TMD-containing PAAR effectors that might require Eag chaperones is also challenging because many PAAR domain-containing effectors lack TMDs (*Shneider et al., 2013*), and aside from being comprised of hydrophobic residues, the TMDs themselves are poorly conserved.

In an attempt to overcome the challenges associated with identifying Eag-interacting T6SS effectors, we used *jackhmmer* to generate a sequence alignment hidden Markov model (HMM) for the N-terminal 60 residues of Tse6 using an iterative search procedure that queried the UniProtKB database (*Johnson et al., 2010*). We reasoned that if there exists a molecular signature present in effector proteins indicative of Eag chaperone association, it would be located within this region of Tse6 homologous proteins because it contains a known chaperone-binding site. Remarkably, the HMM we obtained revealed a nearly invariant AARxxDxxxH motif, which in Tse6 is found in the first 15 residues of the protein and is immediately N-terminal to its first TMD (*Figure 1C*). In total, our query identified over 2054 proteins containing this motif (*Supplementary file 1* and *Figure 1—figure supplement 1A*). Among these candidate effectors, our search identified the recently characterized toxins Tre1, Tas1, and DddA as well as many toxins of unknown function indicating that our approach may have identified T6SS effectors with novel biochemical activities (*Ting et al., 2018*; *Ahmad et al., 2019*; *Mok et al., 2020*). Interestingly, prior to any knowledge of PAAR domains or Eag chaperones being involved in T6SS function, Zhang and colleagues noted the existence of this N-terminal motif in PAAR-containing proteins through an informatics analysis of bacterial nucleic acid degrading toxins (*Zhang et al., 2011*). Here, they refer to the motif and its adjacent TMD region as 'prePAARTM' because these sequence elements co-occur with one another and because they are both found N-terminal to PAAR domains. We have chosen to refer to the motif as 'prePAAR' because, as described below, our data indicate it has a function that is distinct from the TMD regions.

Examination of our putative effector sequences revealed that prePAAR is substantially enriched in bacterial genera with characterized T6SSs including *Pseudomonas*, *Burkholderia*, *Salmonella*, *Shigella*, *Escherichia*, *Enterobacter*, *Yersinia*, and *Serratia*. Interestingly, no prePAAR motifs were identified in *Vibrio* despite an abundance of species within this genus possessing highly active bacteria-targeting T6SSs. We next obtained all 56,324 available genomes from NCBI for the abovementioned genera and found that 26,327 genomes encode at least one prePAAR motif. After removing all redundant sequences, 6101 unique prePAAR-containing proteins present across 5584 genomes were used for further analyses (*Supplementary file 2*, List C). In these genomes, we determined that approximately 90% encode a single prePAAR motif, although instances where prePAAR is present up to six times within a single genome were also identified (*Figure 1D*). To determine if these unique proteins are probable TMD-containing T6SS effectors that require Eag chaperones for secretion, we next examined each prePAAR-containing protein and its associated genome for the following three criteria: (1) the existence of an Eag chaperone encoded in the same genome, (2) the presence of a downstream PAAR domain, and (3) predicted TMDs in the first 300 amino acids of the protein (*Krogh et al., 2001*; *Käll et al., 2007*). The location restriction in our TMD search was used in order to exclude C-terminal toxin domains that possess TMDs, which differ from N-terminal translocation TMDs in that they may not require chaperones for secretion (*Mariano et al., 2019*). We searched each genome for Eag proteins using an HMM for DUF1795 and found that 99.5% (5554/5584) of prePAAR-containing genomes also possessed at least one *eag* gene (*Jones et al., 2014*). In approximately 14% of the 5554 genomes analyzed, the number of prePAAR motifs matched the number of Eag homologues. In the remainder of cases, the number of Eag homologous proteins exceeded the number of prePAAR motifs, with a weighted average of 2.5 paralogues per genome. As is the case with *eagT6-tse6* and *eagR1-rhsA*, ~90% of the identified prePAAR-containing effector genes appear directly beside an *eag* gene, whereas the remaining ~10% are found in isolation suggesting that their putative chaperone is encoded elsewhere in the genome. We removed pre-PAAR-containing protein fragments (proteins less than 100 amino acids in length) and further reduced redundancy by clustering sequences with 95% identity. Remarkably, in all but two of the remaining 1166 prePAAR-containing proteins, we identified a PAAR domain, indicating a probable functional relationship between prePAAR and PAAR. The two prePAAR-containing proteins lacking a PAAR domain were either adjacent to a gene encoding a PAAR domain-containing protein or directly

beside T6SS structural genes. Finally, we searched 1166 prePAAR-containing proteins for TMDs and found that all protein sequences contained predicted TMDs with 86% having one region of TMDs and 14% having two regions of TMDs. In sum, our prePAAR-based search procedure identified thousands of candidate effector proteins possessing properties consistent with the requirement for an Eag chaperone for T6SS-dependent export.

To further analyze our collection of prePAAR-containing effectors, we built a phylogenetic tree from 1166 non-redundant effector sequences that represent the diversity present in our collection of sequences (*Figure 1E*). Interestingly, two distinct sizes of proteins emerged from this analysis: large prePAAR effectors that are on average 1196 amino acids in length and small prePAAR effectors comprised of an average of 443 amino acids (*Figure 1E* and *Figure 1—figure supplement 1B*). As noted previously, all effectors contained predicted TMDs; however, large effectors almost exclusively contained a single region of TMDs N-terminal to their PAAR domain, whereas most small effectors contained TMD regions N- and C-terminal to their PAAR domain. To distinguish between these two domain architectures, we hereafter refer to large, single TMD region-containing prePAAR effectors as class I and small, two TMD region-containing prePAAR effectors as class II. Notably, class I effectors also contain numerous YD repeat sequences, which are a hallmark of *rearrangement hotspot* (Rhs) proteins that function to encapsulate secreted toxins (*Figure 1F*; *Busby et al., 2013*). We also found a small subset of these effectors are encoded by two separate ORFs, the first encoding prePAAR-TMD-PAAR and the second encoding a protein containing Rhs repeats and a C-terminal toxin domain (*Figure 1—figure supplement 1C*). Conversely, class II effectors are distinguished by a GxxxxGxxLxGxxxD motif in addition to their second TMD region.

As a first step toward validating our informatics approach for identifying Eag chaperone-effector pairs, we assessed the ability of several newly identified Eag chaperones to interact with the prePAAR-containing effector encoded in the same genome. We previously demonstrated that the class II effector Tse6 interacts with EagT6 and we similarly found that when expressed in *E. coli*, Eag chaperones from *Enterobacter cloacae*, *Salmonella* Typhimurium, *Shigella flexneri*, and *Serratia proteamaculans* co-purified with their predicted cognate effector (*Figure 1G* and *Figure 1—figure supplement 1C*). Collectively, these findings indicate that prePAAR proteins constitute two classes of TMD-containing T6SS effectors and that representative members from both classes interact with Eag chaperones.

## Eag chaperones are specific for cognate prePAAR effectors

We next sought to examine the specificity of Eag chaperones towards prePAAR effectors in a biologically relevant context. To accomplish this, we inspected our list of prePAAR effectors and found that the soil bacterium *Pseudomonas protegens* Pf-5 possesses both a class I and class II effector, encoded by the previously described effector genes *rhsA* and *tne2*, respectively (*Tang et al., 2018*). Furthermore, the genome of this bacterium encodes two putative Eag chaperones, PFL_6095 and PFL_6099, which have 25% sequence identity between them (*Figure 2A*). PFL_6095 is found upstream of *rhsA* and is likely co-transcribed with this effector, whereas PFL_6099 is not found next to either effector gene. To examine the relationship between these genes, we generated strains bearing single deletions in each effector and chaperone gene and conducted intraspecific growth competition assays against *P. protegens* recipient strains lacking the *rhsA-rhsI* or *tne2-tni2* effector-immunity pairs. We noted that protein secretion by the T6SS of *P. protegens* is substantially inhibited by the threonine phosphorylation pathway, so we additionally inactivated the threonine phosphatase encoding gene *pppA* in recipients to induce a 'tit-for-tat' counterattack by wild-type donor cells (*Figure 2—figure supplement 1A–B*; *Mougous et al., 2007*; *Basler et al., 2013*). Consistent with the effector-immunity paradigm for bacteria-targeting T6SSs, wild-type *P. protegens* readily outcompeted Δ*rhsA* Δ*rhsI* Δ*pppA* and Δ*tne2* Δ*tni2* Δ*pppA* strains in a *rhsA*- and *tne2*-dependent manner, respectively (*Figure 2B*). Additionally, we found that a strain lacking PFL_6095 no longer exhibited a co-culture fitness advantage versus a Δ*rhsA* Δ*rhsI* Δ*pppA* recipient but could still outcompete *tne2* sensitive recipients to the same extent as the wild-type strain. Conversely, a ΔPFL_6099 strain outcompeted Δ*rhsA* Δ*rhsI* Δ*pppA* but not Δ*tne2* Δ*tni2* Δ*pppA* recipients. Together, these data indicate that the delivery of RhsA and Tne2 into susceptible target cells requires effector-specific *eag* genes.

To test the ability of PFL_6095 and PFL_6099 to act as RhsA- and Tne2-specific chaperones, respectively, we co-expressed each chaperone-effector pair in *E. coli* and examined intracellular

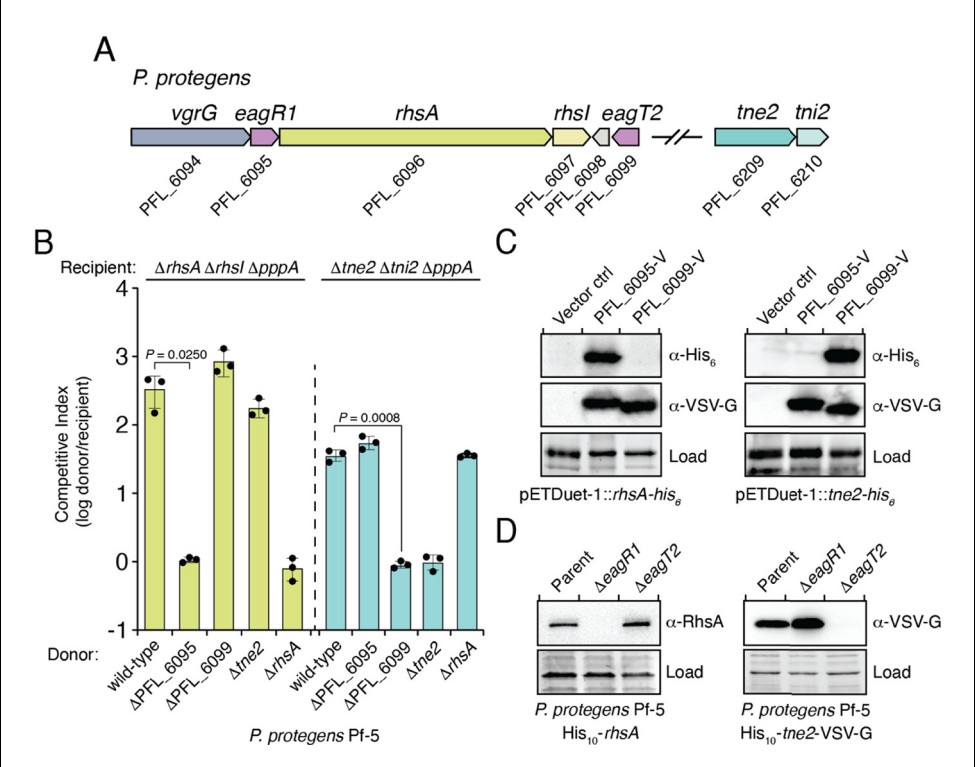

**Figure 2.** Eag chaperones are specific for their cognate prePAAR effector and are necessary for effector stability in vivo. (**A**) Genomic context of two prePAAR-containing effector-immunity pairs from *P. protegens* Pf-5. RhsA is a class I effector (shown in green) and Tne2 is a class II effector (shown in blue). Shading is used to differentiate effector (dark) and immunity genes (light). Predicted *eag* genes are shown in purple. (**B**) Outcome of intraspecific growth competition assays between the indicated *P. protegens* donor and recipient strains. Donor strains were competed with recipient strains lacking *rhsA-rhsI* (green) or *tne2-tni2* (blue). Both recipients are lacking *pppA* to stimulate type VI secretion. Data are mean ± s.d. for *n* = 3 biological replicates and are representative of two independent experiments; p values shown are from two-tailed, unpaired *t*-tests. (**C**) Western blot analysis of *E. coli* cell lysates from cells expressing the indicated effectors (RhsA or Tne2) and either empty vector, PFL_6095 V or PFL_6099 V. (**D**) Affinity-tagged RhsA or Tne2 were purified from cell fractions of the indicated *P. protegens* strains and visualized using western blot analysis. Deletion constructs for each *eag* gene were introduced into each of the indicated parent backgrounds. A non-specific band present in the SDS-PAGE gel was used as a loading control. (**C–D**) Data are representative of two independent experiments.

The online version of this article includes the following figure supplement(s) for figure 2:

**Figure supplement 1.** The type VI secretion system of *P. protegens* Pf-5 is repressed by the threonine phosphorylation pathway.

effector levels by western blot. Consistent with functioning to promote cognate effector stability, accumulation of RhsA only occurred in the presence of PFL_6095, whereas Tne2 accumulated in cells containing PFL_6099 (**Figure 2C**). We next examined the stability-enhancing properties of PFL_6095 and PFL_6099 when expressed at native levels in *P. protegens*. Due to challenges associated with detecting RhsA and Tne2 in unconcentrated cell lysates, we constructed chromosomally encoded N-terminal decahistidine-tagged (his$_{10}$) fusions of RhsA and Tne2 to facilitate the enrichment of these proteins from *P. protegens* and confirmed that these fusions did not compromise the ability of these effectors to intoxicate recipients (**Figure 3—figure supplement 1A–B**). Following affinity purification, RhsA and Tne2 levels were assessed using RhsA and vesicular stomatitis virus glycoprotein epitope (VSV-G) antibodies, respectively. In line with our data in *E. coli*, we were unable to detect RhsA in the absence of PFL_6095 whereas Tne2 was absent in a strain lacking PFL_6099 (**Figure 2D**). Collectively, these data suggest that Eag chaperones exhibit a high degree of specificity for their cognate effectors. Based on our characterization of these genes, we propose to rename PFL_6095

and PFL_6099 to *eagR1* and *eagT2*, respectively, to reflect their newfound role as chaperones for the prePAAR-containing effectors RhsA and Tne2.

Previous biochemical studies on the class II prePAAR effector Tse6 and its cognate chaperone EagT6 demonstrated that the two TMD regions of this effector each require an EagT6 chaperone for stability (*Quentin et al., 2018*). These findings suggest that there may exist a physical limitation to the number of TMDs that a single EagT6 chaperone can stabilize. Our finding that class I prePAAR effectors contain only one TMD region suggests that these proteins may only possess one Eag inter-action site (*Figure 3A*). To test this, we constructed a RhsA variant lacking its N-terminal region (RhsA$_{\Delta NT}$) and co-expressed this truncated protein with EagR1 in *E. coli*. Consistent with our hypothesis, affinity purification of RhsA$_{\Delta NT}$ showed that this truncated variant does not co-purify with EagR1 (*Figure 3B*). Additionally, expression of the 74 residue N-terminal fragment of RhsA in isola-tion was sufficient for EagR1 binding (*Figure 3—figure supplement 1C*). Our data also demonstrate that in contrast to wild-type RhsA, RhsA$_{\Delta NT}$ is stable in the absence of EagR1 when expressed in *E. coli* indicating that the N-terminus imparts instability on the protein in the absence of its cognate chaperone. In *P. protegens*, we could readily detect *rhsA*$_{\Delta NT}$ in a strain lacking *eagR1*, corroborating our findings in *E. coli* (*Figure 3C*). However, despite the enhanced stability of chaperone 'blind' RhsA$_{\Delta NT}$, a *P. protegens* strain expressing this truncation was no longer able to outcompete RhsA-

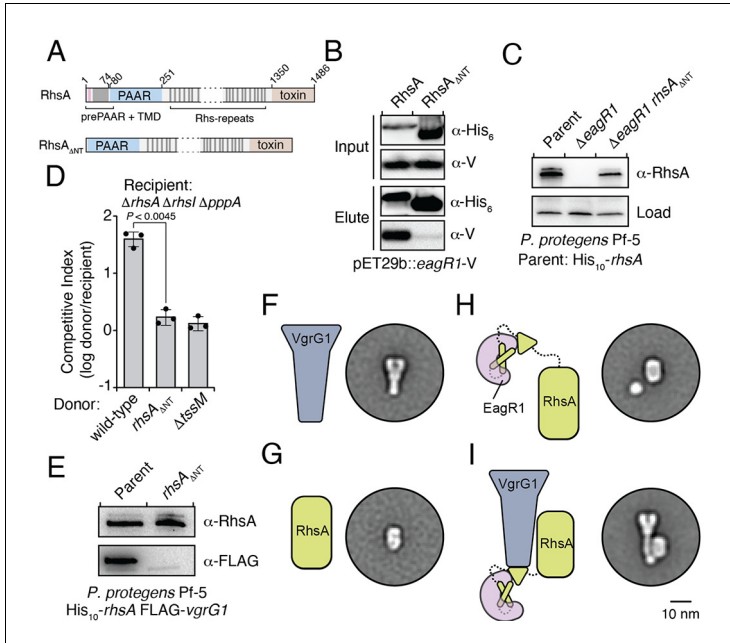

**Figure 3.** An Eag chaperone promotes the stability of its cognate class I prePAAR effector by interacting with its prePAAR and TMD-containing N-terminus. (**A**) Domain architecture of *P. protegens* RhsA and a truncated variant lacking its prePAAR and TMD-containing N-terminus (RhsA$_{\Delta NT}$). (**B**) EagR1 interacts with the N-terminus of RhsA. His$_6$-tagged RhsA or RhsA$_{\Delta NT}$ and co-expressed with EagR1 in *E. coli*, purified using affinity chromatography and detected by western blot. (**C**) Affinity purification of chromosomally His$_{10}$-tagged RhsA or RhsA$_{\Delta NT}$ from cell fractions of the indicated *P. protegens* strains. The parent strain expresses chromosomally encoded His$_{10}$-tagged RhsA. The loading control is a non-specific band on the blot. (**D**) Outcome of growth competition assays between the indicated donor and recipient strains of *P. protegens*. Data are mean ± s.d. for *n* = 3 biological replicates; p value shown is from a two-tailed, unpaired *t*-test. (**E**) Affinity purification of His$_{10}$-RhsA or His$_{10}$-RhsA$_{\Delta NT}$ from a *P. protegens* Pf-5 strain containing a chromosomally encoded FLAG epitope tag fused to *vgrG1*. FLAG-tagged VgrG1 was detected by western blot. (**F–I**) Representative negative-stain EM class averages for purified VgrG1 (**F**), RhsA$_{\Delta NT}$ (**G**), EagR1-RhsA complex (**H**) and EagR1-RhsA-VgrG complex (**I**). Scale bar represents 10 nm for all images. All proteins were expressed and purified from *E. coli*. (**B–C, E**) Data are representative of two independent experiments.

The online version of this article includes the following figure supplement(s) for figure 3:

**Figure supplement 1.** RhsA interacts with EagR1 and requires VgrG1 for delivery into target cells.

sensitive recipient cells demonstrating an essential role for the chaperone-bound N-terminus during interbacterial competition (*Figure 3D*).

After ruling out the possibility that truncating the N-terminus of RhsA affects the growth-inhibitory activity of its C-terminal toxin domain (*Figure 3—figure supplement 1D*), we next examined the ability of RhsA$_{\Delta NT}$ to interact with its cognate secreted structural component of the T6SS apparatus. T6SS effectors encoded downstream of *vgrG* genes typically rely on the encoded VgrG protein for delivery into target cells (*Whitney et al., 2014*). Consistent with this pattern, PFL_6094 encodes a predicted VgrG protein, herein named VgrG1, which we confirmed is required for RhsA-mediated growth inhibition of susceptible target cells (*Figure 3—figure supplement 1E*). Furthermore, using a *P. protegens* strain expressing His$_{10}$-tagged RhsA and FLAG-tagged VgrG1 from their native loci, we found that these proteins physically interact to form a complex (*Figure 3E*). To test if the absence of the chaperone-bound N-terminus affects the formation of this complex, we used our *E. coli* co-expression system to purify RhsA-EagR1-VgrG1 complexes. These experiments show that RhsA$_{\Delta NT}$ is not able to interact with VgrG1, even though this truncated protein possesses its PAAR domain, which in T6SS effectors lacking prePAAR and TMDs in their N-terminus, is sufficient for VgrG interaction (*Figure 3—figure supplement 1F*; *Bondage et al., 2016*).

To gain insight into how EagR1 binding facilitates RhsA interaction with VgrG1, we next performed negative-stain electron microscopy (EM) to examine the configuration of each subunit within this complex. To facilitate the accurate identification of each component, we obtained class averages of purified VgrG1, RhsA$_{\Delta NT}$, RhsA-EagR1 complex, and RhsA-EagR1-VgrG1 complex (*Figure 4—figure supplement 1A–H*). As expected, isolated VgrG1 and RhsA$_{\Delta NT}$ proteins appeared as characteristic spike- and barrel-shaped proteins, respectively (*Spínola-Amilibia et al., 2016*; *Busby et al., 2013*; *Figure 3F and G*). Intriguingly, images of RhsA-EagR1 complexes contained a sphere-shaped object that likely represents a subcomplex between EagR1 and the N-terminus of RhsA (*Figure 3H*). Lastly, the class-averages of RhsA-EagR1-VgrG1 complexes revealed a close association of EagR1 and RhsA with the tip of the VgrG spike, which is likely mediated by the PAAR domain of RhsA (*Figure 3I*). Interestingly, although both complexes exhibit significant rotational flexibility, the average distance between the subcomplex formed by EagR1 and the N-terminus of RhsA is substantially greater in the absence of VgrG1 (average distance: 2.68 nm, *n* = 27 classes versus 1.20 nm, *n* = 26 classes) (*Figure 4—figure supplement 1F–H*). When taken together with our biochemical experiments, these structural data indicate that EagR1 stabilizes the N-terminus of RhsA, which may also orient the effector such that it can interact with its cognate VgrG.

## Eag chaperones bind effector TMDs by mimicking transmembrane helical packing

In addition to a TMD-containing region, the N-terminus of prePAAR effectors also harbors the prePAAR motif itself. However, the negative stain EM images of RhsA-EagR1-VgrG1 particles presented herein and our previously determined single-particle cryo-EM structure of a complex containing Tse6-EagT6-VgrG1 are of insufficient resolution to resolve the structures of chaperone-bound effector TMDs or the prePAAR motif (*Quentin et al., 2018*). Therefore, to better understand the molecular basis for chaperone-TMD interactions and to gain insight into prePAAR function, we initiated X-ray crystallographic studies on both class I and class II effector-chaperone complexes. Efforts to co-crystallize *P. protegens* EagR1 with the prePAAR and TMD-containing N-terminus of RhsA were unsuccessful. However, the EagR1 homologue SciW from *Salmonella* Typhimurium crystallized in isolation and in the presence of the N-terminus of the class I prePAAR effector Rhs1 (Rhs1$_{NT}$), allowing us to determine apo and effector-bound structures to resolutions of 1.75 Å and 1.90 Å, respectively (*Figure 4* and *Table 1*). Similar to RhsA, we confirmed that a Rhs1$_{\Delta NT}$ variant was unable to bind its cognate chaperone, SciW (*Figure 3—figure supplement 1G*). The structure of the EagT6 chaperone was previously solved as part of a structural genomics effort and we were additionally able to obtain a 2.60 Å co-crystal structure of this chaperone in complex with the N-terminal prePAAR and first TMD region of the class II effector Tse6 (Tse6$_{NT}$) (*Figure 4* and *Table 1*).

The overall structure of SciW reveals a domain-swapped dimeric architecture that is similar to the previously described apo structure of *P. aeruginosa* EagT6, although each chaperone differs in its electrostatic surface properties (*Figure 5—figure supplement 1A–D*; *Whitney et al., 2015*). A comparison of the chaperone structures in their apo and effector-bound states shows that upon effector binding, both chaperones 'grip' the prePAAR-TMD region of their cognate effector in a claw-like

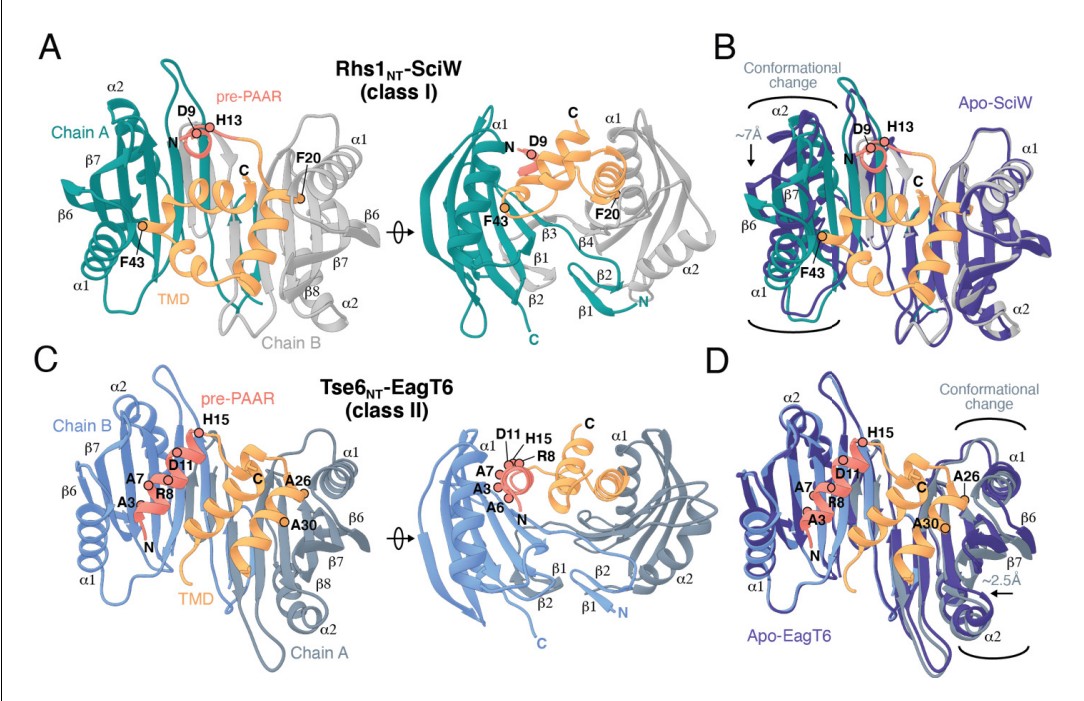

**Figure 4.** Co-crystal structures of the N-terminus of class I and class II prePAAR effectors in complex with their cognate Eag chaperones. (A) An X-ray crystal structure of the Eag chaperone SciW bound to the N-terminus of *Salmonella* Typhimurium class I prePAAR effector Rhs1 (Rhs1$_{NT}$, residues 8–57 are modeled) shown in two views related by a ~ 90° rotation. (B) Structural overlay of the apo-SciW structure with SciW-Rhs1$_{NT}$ complex demonstrates that a considerable conformational change in SciW occurs upon effector binding. (C) An X-ray crystal structure of the Eag chaperone EagT6 bound to the N-terminus of Tse6 (Tse6$_{NT}$, residues 1–38 and 41–58 are modeled) shown in two views related by a ~ 90° rotation. (D) Structural overlay of the apo-EagT6 structure (PDB 1TU1) with the EagT6-Tse6$_{NT}$ complex shows a minor conformational change in EagT6 occurring upon effector binding. Eag chaperones are colored by chain, N-terminal transmembrane domains (TMDs) are colored in orange, the pre-PAAR motif in red, and apo chaperone structure in dark blue. Positions of residues of interest in the effector N-terminal regions are labeled.

The online version of this article includes the following figure supplement(s) for figure 4:

**Figure supplement 1.** RhsA, EagR1, and VgrG1 form a ternary complex in vitro.

manner (*Figure 4A–D*). Although our biochemical data indicate that Eag chaperones exhibit a high degree of specificity for their associated effector, the internal surface of the claw-shaped dimer contains a number of conserved residues that make critical interactions with the TM helices in both complexes (*Figure 5A–F*). For example, I22 and I24 of EagT6 create a hydrophobic surface in the 'palm' of the claw, which is flanked on either side by symmetrical hydrophobic surfaces comprised of A62, L66, L98, F104, and I113 (*Figure 5B–D*). Furthermore, the conserved hydrophilic residues S37, S41, Q58, and Q102 also interact with the bound effectors by making bifurcated hydrogen bonds to amide or carbonyl groups in the peptide backbone of the TM helices (*Figure 5E–F*). These polar interactions between chaperone and effector TM helices are striking because they are reminiscent of polar interactions seen within the helical packing of alpha helical transmembrane proteins, which often use serine and glutamine residues to mediate inter-helical interactions via bifurcated hydrogen bonds between side chain and main chain atoms (*Dawson et al., 2002*; *Dawson et al., 2003*; *Adamian and Liang, 2002*). Additionally, EagT6 and SciW provide 'knob-hole-like' interactions, which also feature prominently in membrane protein packing (*Curran and Engelman, 2003*). Knob-hole interactions involve a large hydrophobic residue on one TM helix acting as a 'knob' to fill the hole provide by a small residue such as glycine or alanine on another TM-helix. TM holes are typically created by GxxxG/A motifs such as those found in G19-A24 (Rhs1) and G25-A30 (Tse6). In this case, the conserved Eag chaperone residue L66 provides a knob for the A24/30 hole (*Figure 5E–F*). Given that the Eag chaperone dimer creates a hydrophobic environment with complementary knob-hole interactions for its cognate effector TM helices, and interacts with TM helices via side chain to main chain hydrogen bonds, we conclude that Eag chaperones provide an environment that mimics

**Table 1.** X-ray data collection and refinement statistics.

| | SciW (native) | SciW (Iodide) | SciW-Rhs1$_{1-59}$ | EagT6-Tse6$_{1-61}$ |
|---|---|---|---|---|
| **Data Collection** | | | | |
| Wavelength (Å) | 1.5418 | 1.5418 | 0.97895 | 1.5418 |
| Space group | P2$_1$2$_1$2$_1$ | P2$_1$2$_1$2$_1$ | P3$_1$21 | P3$_2$ |
| **Cell dimensions** | | | | |
| $a, b, c$ (Å) | 55.27 75.1 76.6 | 55.6 75.3 76.4 | 105.3 105.3 248.4 | 68.9 68.9 173.1 |
| $\alpha, \beta, \gamma$ (°) | 90 90 90 | 90 90 90 | 90 90 120 | 90 90 120 |
| Resolution (Å) | 29.03–1.75 | 19.63–2.21 | 91.20–1.90 | 28.22–2.55 |
| | (1.82–1.75) | (2.33–2.21) | (1.98–1.90) | (2.65–2.55) |
| Unique reflections | 32309 (3162)* | 29933 (4888) | 126298 (12473) | 29267 (2832) |
| CC(1/2) | 99.8 (89.1) | 99.6 (81.4) | 99.9 (53.9) | 99.6 (52.8) |
| $R_{merge}$ (%)[†] | 6.2 (91.3) | 6.1 (44.7) | 5.7 (34.6) | 15.5 (179.8) |
| $I/\sigma I$ | 14.2 (1.9) | 8.0 (1.8) | 11.6 (1.26) | 7.27 (0.92) |
| Completeness (%) | 99.5 (98.8) | 96.0 (97.9) | 99.9 (99.9) | 99.3 (96.9) |
| Redundancy | 7.0 (6.8) | 2.0 (1.9) | 9.9 (9.7) | 4.9 (4.8) |
| **Refinement** | | | | |
| $R_{work}/R_{free}$ (%)[‡] | 19.8/22.6 | | 18.7/21.4 | 22.9/26.6 |
| Average B-factors (Å$^2$) | 46.1 | | 42.9 | 71.7 |
| Protein | 45.1 | | 42.5 | 72.1 |
| Ligands | 60.8 | | 123.4 | |
| Water | 53.9 | | 42.2 | 59.3 |
| **No. atoms** | | | | |
| Protein | 2331 | | 10492 | 7827 |
| Ligands | 10 | | 60 | |
| Water | 256 | | 1119 | 248 |
| **Rms deviations** | | | | |
| Bond lengths (Å) | 0.003 | | 0.005 | 0.004 |
| Bond angles (°) | 0.67 | | 0.68 | 0.73 |
| **Ramachandran plot (%)[§]** | | | | |
| Total favored | 99.65 | | 99.24 | 98.26 |
| Total allowed | 0.35 | | 0.68 | 1.74 |
| PDB code | 6XRB | | 6XRR | 6XRF |

*Values in parentheses correspond to the highest resolution shell.

[†]$R_{merge} = \Sigma \, \Sigma \, |I(k) - <I>| / \Sigma \, I(k)$ where $I(k)$ and $<I>$ represent the diffraction intensity values of the individual measurements and the corresponding mean values. The summation is over all unique measurements.

[‡]$R_{work} = \Sigma \, ||F_{obs}| - k|F_{calc}||/|F_{obs}|$ where $F_{obs}$ and $F_{calc}$ are the observed and calculated structure factors, respectively. $R_{free}$ is the sum extended over a subset of reflections excluded from all stages of the refinement.

[§]As calculated using MOLPROBITY (**Chen et al., 2010**).

transmembrane helical packing to stabilize prePAAR effector TMDs in the cytoplasm prior to effector export from the cell.

## prePAAR facilitates PAAR domain interaction with the VgrG spike

We next compared the conformation of the bound prePAAR-TMD fragments between our effector-chaperone co-crystal structures. Interestingly, despite the abovementioned similarities between the SciW and EagT6 structures, the conformation of the N-terminal fragment of their bound prePAAR effector differs significantly. In the SciW complex, Rhs1$_{NT}$ adopts an asymmetric binding mode

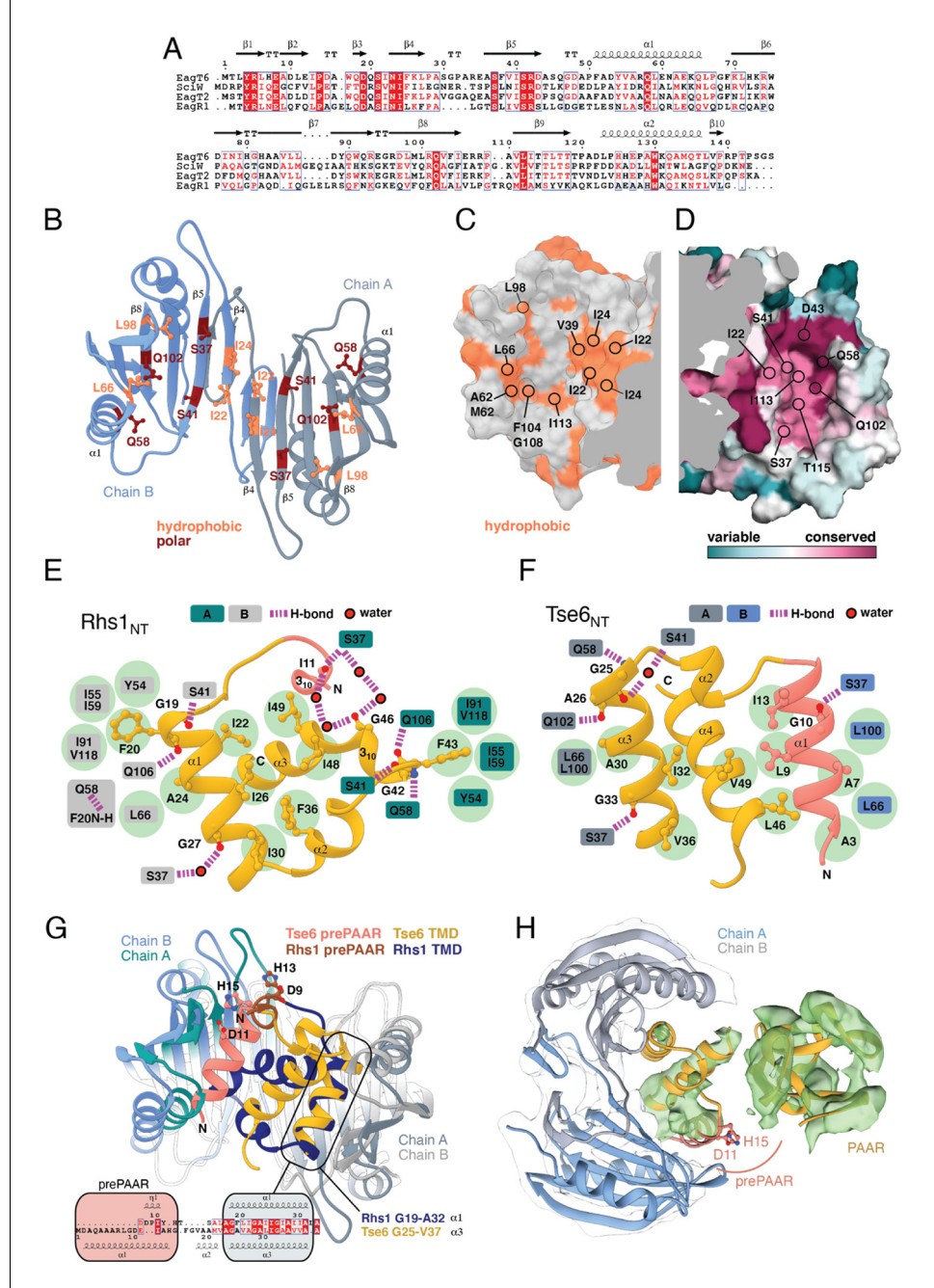

**Figure 5.** Eag chaperones interact with effector TMDs by mimicking interhelical interactions of alpha helical membrane proteins. (**A**) Alignment of Eag chaperones that interact with class I (SciW, EagR1) or class II (EagT6 and EagT2) prePAAR effectors plotted with secondary structure elements. (**B**) Residues making intimate molecular contacts with their respective TMDs that are conserved among SciW, EagR1, EagT6, and EagT2 are shown. Hydrophobic contacts are colored in light orange and polar contacts in deep red. Residue numbers are based on EagT6. (**C and D**) The conserved hydrophobic molecular surface of the chaperones is shown in light orange (**C**) and their molecular surface residue conservation is shown as determined by the Consurf server (**D**) (*Ashkenazy et al., 2016*). Conserved residues making contacts with the TMDs in both co-crystal structures are shown. (**E**) Molecular contact map of Rhs1NT (residues 1–59) and SciW. prePAAR is shown in pink and the TMD regions in gold. Amino acids making contacts with the conserved residues of the Eag chaperones are shown by side chain/and or by main chain atoms (red for oxygen and blue for nitrogen). Residues in the Eag chaperone are highlighted by color of chain A or B. Polar (H-bond) contacts are drawn with a purple dashed line and are made with the side chain of the listed Eag residue. Outlined red circles indicate a water molecule. Light green circles

*Figure 5 continued on next page*

*Figure 5 continued*

indicate van der Waals interactions and hydrophobic interactions. The central group of hydrophobic residues without a listed chaperone residue all pack into the Eag hydrophobic face in *Figure 4G* (EagT6 I22/24 and V39). (F) Molecular contact map of Tse6$_{NT}$ (residues 1–61) and EagT6. Schematic is the same as panel B. Q102 in EagT6 corresponds to Q106 in SciW. (G) Structural alignment of SciW-Rhs1$_{NT}$ and EagT6-Tse6$_{NT}$ co-crystal structures using the structurally conserved TM helix as a reference. Eag chain coloring is the same as *Figure 4*. Rhs1$_{NT}$ is colored in dark blue with a brown prePAAR and Tse6$_{NT}$ in gold with a pink prePAAR. The conserved solvent accessible prePAAR residues D9/11 and H13/15 are shown in ball and stick model. Inset sequence alignment reflects the structurally aligned residues of Rhs1$_{NT}$ (top) and Tse6$_{NT}$ (bottom) as calculated by UCSF Chimera (*Pettersen et al., 2004*). Secondary structural elements are labeled. (H) Docking of the EagT6-TMD crystal structure from *Figure 4C* into the previously obtained cryo-EM density map of the EagT6-Tse6-EF-Tu-Tsi6-VgrG1a complex (EMD-0135). Cryo-EM density corresponding to EagT6 is depicted in transparent gray and Tse6-TMD and Tse6-PAAR in green; prePAAR residues are shown in pink. Note that Tse6-TMD was docked independent of EagT6 into the Tse6 density. One of three possible orientations for the PAAR domain is shown.

The online version of this article includes the following figure supplement(s) for figure 5:

**Figure supplement 1.** Structural comparison of Eag chaperones and effector complexes.

whereby the effector fragment does not make equivalent molecular contacts with both chains of the two-fold symmetrical chaperone dimer (*Figures 4A* and *5F*). The first TM helix (residues 19–33) binds to the hydrophobic cavity of one SciW protomer whereas the remaining hydrophobic region of Rhs1, which consists of two anti-parallel alpha-helices connected by a short 3$_{10}$ helix, occupies the remainder of the binding surface. Phenylalanine residues F20 and F43 likely play an important role in the asymmetric binding of Rhs1 to SciW because their hydrophobic side chains insert into equivalent hydrophobic pockets found in each SciW protomer (*Figure 5E*). By contrast, Tse6$_{NT}$ exhibits a pseudosymmetric binding mode with EagT6 (*Figures 4C* and *5F*). In this structure, two alpha-helices of Tse6 each occupy equivalent Eag binding pockets and run in the opposite direction to match the antiparallel arrangement of the EagT6 dimer. For example, A7 and A30 of Tse6 interact with equivalent sites in their respective chaperone protomers (*Figures 4B* and *5E*). These two helices, which consist of prePAAR and a TM helix, flank a central TM helix whose C-terminus extends into the solvent, likely indicating the location of the downstream PAAR domain in the full-length effector.

A lack of interpretable electron density prevented modelling of Rhs1's entire AARxxDxxxH prePAAR motif in our Rhs1$_{NT}$-SciW co-crystal structure. However, the DxxxH portion of this motif is part of a short 3$_{10}$ helix that orients the aspartate and histidine side chains such that they face outward into solvent (*Figure 5—figure supplement 1E–G*). By contrast, we were able to model the entire prePAAR motif of Tse6$_{NT}$ and in this case, the motif forms an alpha helix that binds the hydrophobic pocket of an EagT6 protomer. In this structure, the two conserved alanine residues of prePAAR make contact with the EagT6 chaperone, whereas the arginine, aspartate and histidine residues are solvent exposed (*Figures 4C* and *5F*). Remarkably, despite existing in different secondary structure elements, the D11 and H15 prePAAR residues of Tse6 are located in a similar 3D location as their D9 and H13 counterparts in Rhs1 (*Figure 5G*). It should be noted that the modelled conformation of Tse6$_{NT}$ appears to be locked into place by crystal packing, suggesting that in solution Tse6's prePAAR motif may exhibit significant conformational flexibility and can dissociate from EagT6 as is observed for the prePAAR motif of Rhs1 (*Figure 5—figure supplement 1H–I*). In support of this, we previously showed that addition of detergent disrupts the interaction between EagT6 and Tse6 suggesting that Eag chaperone-effector interactions are labile, likely because chaperone dissociation is required prior to effector delivery into target cells (*Quentin et al., 2018*). Intriguingly, docking our high-resolution EagT6-Tse6$_{NT}$ crystal structure into our previously determined lower resolution Tse6-EagT6-VgrG1 cryo-EM map orients the D11 and H15 prePAAR residues of Tse6 in a position that suggests they interact with its PAAR domain (*Figure 5H*). In sum, our structural analyses of prePAAR shows that this region is likely dynamic, and its mode of interaction varies for class I and class II prePAAR effectors. However, both Eag chaperones bind the N-terminus of their cognate effector such that the conserved aspartate and histidine residues of prePAAR are positioned to potentially be involved in interactions with the downstream PAAR domain, and thus may play a role in effector-VgrG interactions.

To test if prePAAR influences PAAR function, we next conducted mutagenesis analysis on Tse6 because its PAAR-dependent interaction with its cognate VgrG protein, VgrG1a, can be monitored in vivo by western blot. During denaturing electrophoresis, Tse6 appears in two forms: (1) a high-molecular-weight species corresponding to Tse6-VgrG1a complex and (2) a low-molecular-weight species indicative of free Tse6 (*Whitney et al., 2015*). Deletion of *vgrG1a* only affects complex formation, whereas deletion of the *eagT6* gene results in a substantial reduction in the levels of both species providing a means to differentiate residues involved in effector-chaperone versus effector-VgrG interactions (*Quentin et al., 2018*). Using this readout, we engineered *P. aeruginosa* strains expressing Tse6 D11A and H15A single amino acid substitutions and a D11A/H15A double substitution and examined the consequences of these prePAAR mutations on Tse6 interactions. In support of a role in promoting proper folding of PAAR, Tse6-VgrG1a complex formation was substantially reduced in a strain expressing the Tse6$^{D11A}$ variant and abolished in a strain expressing Tse6$^{D11A, H15A}$ (*Figure 6A*). We next examined the effect of these mutations on T6SS-dependent delivery of Tse6 into target cells by subjecting these *P. aeruginosa* strains to growth competition assays against Tse6-sensitive recipients. In agreement with our biochemical data, strains expressing Tse6 harboring

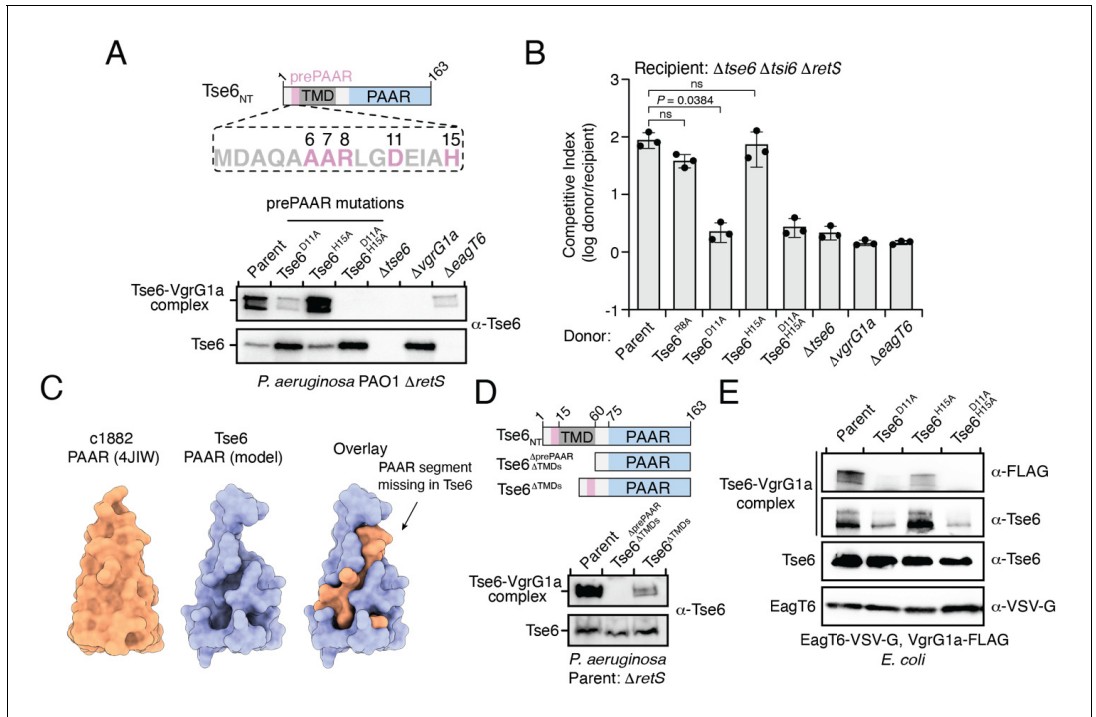

**Figure 6.** prePAAR is required for PAAR domain interaction with the VgrG spike. (A) Western blot analysis of Tse6 from cell fractions of the indicated *P. aeruginosa* strains. Low-molecular-weight band indicates Tse6 alone whereas high-molecular-weight band indicates Tse6-VgrG1a complex. The parental strain contains a Δ*retS* deletion to transcriptionally activate the T6SS (*Goodman et al., 2004*). Schematic shows the N-terminal construct of Tse6 (Tse6$_{NT}$), prePAAR is indicated in pink. (B) Outcome of growth competition assay between the indicated *P. aeruginosa* donor and recipient strains. The parent strain is *P. aeruginosa* Δ*retS*. Data are mean ± s.d. for *n* = 3 biological replicates; p value shown is from a two-tailed, unpaired *t*-test; ns indicates data that are not significantly different. (C) Structural comparison of the c1882 PAAR protein from *E. coli* (PDB: 4JIW) with a model of the PAAR domain of Tse6 generated using Phyre$^2$ (*Kelley et al., 2015*). The overlay shows the additional N-terminal segment present in c1882 that is absent in Tse6. (D) Western blot of cell fractions of the indicated *P. aeruginosa* strains. The identity of the low-molecular weight and high-molecular weight bands are the same as described in A. The parent strain contains a *retS* deletion. Schematic shows mutants of Tse6 natively expressed in *P. aeruginosa*. Only the N-terminus of Tse6 is shown for simplicity. prePAAR is indicated in pink. Constructs lacking TMD1 also lack TMD2. (E) Western blot of elution samples from affinity pull-down with the indicated His$_6$-tagged Tse6 variants co-purified with VgrG1a-FLAG and EagT6-VSV-G in *E. coli*. (A, D–E) Data are representative of two independent experiments.

The online version of this article includes the following figure supplement(s) for figure 6:

**Figure supplement 1.** The PAAR domain of prePAAR effectors lacks a critical N-terminal segment.
**Figure supplement 2.** Orphan PAARs are ancestral to split PAARs.

a D11A mutation exhibited a substantial reduction in co-culture fitness consistent with an inability of these mutant proteins to form a complex with VgrG1a (*Figure 6B*).

To better understand why Tse6's PAAR domain requires prePAAR for function, we compared its sequence and predicted structure to the X-ray crystal structure of the 'orphan' PAAR domain c1882 from *E. coli*, which does not contain additional components such as TMDs or a toxin domain (*Shneider et al., 2013*). Interestingly, this analysis suggested that the PAAR domain of Tse6 lacks an N-terminal segment, which, based on the structure of c1882, is potentially important for the proper folding of this domain (*Figure 6C*). We next extended this structural analysis to include all PAAR domains of the prePAAR effectors that we experimentally confirmed bind Eag chaperones. In all cases, the N-terminal segment of each PAAR domain was missing (*Figure 6—figure supplement 1A*). We also noted that the prePAAR motif possesses significant sequence homology to the N-terminal segment of c1882, suggesting that even though this stretch of amino acids exists on the opposite side of the first TMD region of Tse6, it may comprise the missing segment of Tse6's PAAR domain (*Figure 6—figure supplement 1B*). Lending further support to this hypothesis, when we artificially fused Tse6's prePAAR motif (residues 1–16) with its PAAR domain (residues 77–163) and generated a structural model, we found that the first 16 residues of Tse6 fill the missing structural elements of Tse6's PAAR domain (*Figure 6—figure supplement 1C*). Based on this information, we hypothesized that prePAAR is necessary for PAAR domain function and thus, facilitates VgrG binding. To explore this experimentally, we assessed the effect of prePAAR on VgrG binding in vivo and in vitro. We started by generating two Tse6 mutants expressed from their native locus in *P. aeruginosa*. The first mutant lacks prePAAR, TMD1, and TMD2 (Tse6$^{\Delta prePAAR,\ \Delta TMDs}$), while the second contains prePAAR fused to PAAR and thus lacks only its TMDs (Tse6$^{\Delta TMDs}$). We next assessed the ability of these truncated forms of Tse6 to form a complex with VgrG1a in vivo, as described above. In these experiments, we found that prePAAR and PAAR are together both necessary and sufficient for the formation of Tse6-VgrG1a complexes in vivo (*Figure 6D*). Of note, the amount of the complex formed by the Tse6$^{\Delta TMDs}$ mutant is less than the parent strain, which may be due to unstable structural elements that arose from suboptimal boundaries selected for truncating the effector. We next assessed the formation of this complex in vitro and found that co-incubation of Tse6, EagT6 and VgrG1a after overexpression in *E. coli* leads to the formation of SDS-resistant Tse6-VgrG1a complexes, whereas doing so with a strain expressing Tse6$^{D11A,\ H15A}$ does not (*Figure 6E* and *Figure 6—figure supplement 1F*). Importantly, these mutations do not affect overall levels of Tse6 in cells or affect its ability to bind to EagT6, indicating that these mutations do not have a global destabilizing effect on Tse6 (*Figure 6E*). Together, these data suggest that the PAAR domains of prePAAR effectors exist as 'split PAAR' due to the presence of N-terminal TMDs.

In orphan PAAR proteins, such as c1882, DxxxH motifs are necessary for $Zn^{2+}$-coordination and are therefore necessary for proper folding of this domain (*Shneider et al., 2013*). In agreement with this precedent, the conserved histidine residue in the DxxxH portion of Tse6's prePAAR motif is predicted to be in the same 3D position as the first zinc-coordinating histidine residue of c1882 (*Figure 6—figure supplement 1D*). To extend this comparison further, we conducted in silico analyses to examine the evolutionary relationship and potential $Zn^{2+}$-binding residues in 564 orphan PAARs and 1765 prePAAR effectors. We found that orphan PAAR sequences are ancestral to split PAAR domains and that while orphan PAAR proteins typically contain a total of four histidine and/or cysteine $Zn^{2+}$-coordinating residues, prePAAR effectors only contain three in their PAAR domain with the fourth likely being provided by the prePAAR motif (*Figure 6—figure supplement 1E* and *Figure 6—figure supplement 2*). In support of this prediction, we found that Tse6-VgrG1a complexes formed by the D11A or H15A variants were susceptible to heat treatment under denaturing conditions whereas the wild-type complex remained intact (*Figure 6—figure supplement 1F–G*). Collectively, our experimental data and informatics analyses indicate that unlike orphan PAAR proteins, which contain all the necessary molecular determinants for proper folding, prePAAR effectors may contain inherently unstable PAAR domains that require a prePAAR motif to ensure their proper folding thus enabling their interaction with their cognate VgrG protein (*Figure 7*).

## Discussion

Protein secretion systems endow bacteria with a significant fitness advantage in their niche (*Galán and Waksman, 2018*). The proper functioning of these pathways requires the precise

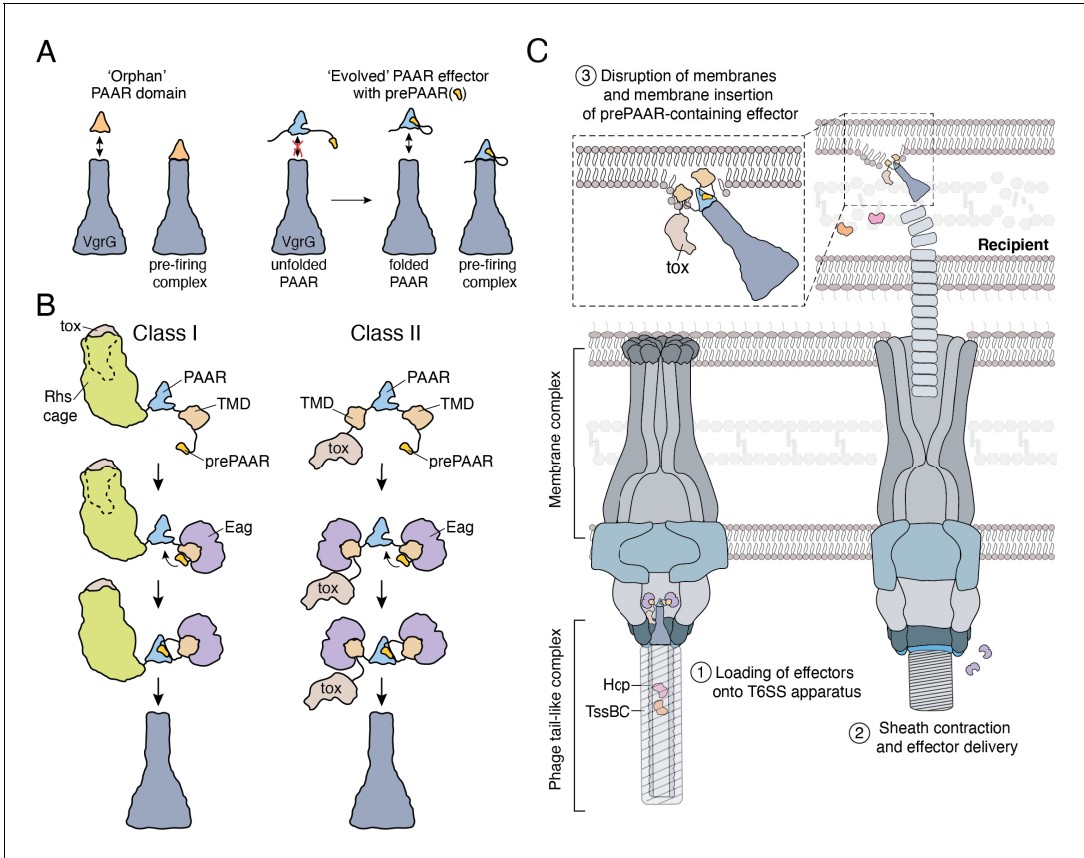

**Figure 7.** Model depicting the role of Eag chaperones and prePAAR in type VI secretion. (**A**) PAAR proteins exist with or without prePAAR domains. Those that lack prePAAR (orphan), can interact with VgrG and form a functional T6SS spike complex without any additional factors. By contrast, prePAAR-containing effectors contain multiple domains (evolved) and likely require the prePAAR motif for proper folding of the PAAR domain and thus, loading onto the T6SS apparatus. (**B**) prePAAR-containing effectors can be divided into two classes: class I effectors have a single TMD and contain a C-terminal toxin domain that is likely housed within a Rhs cage whereas class II effectors contain two TMDs and do not possess a Rhs cage. TMD-chaperone and prePAAR-PAAR interactions are required for effector stability and VgrG interaction, respectively, for both classes of prePAAR effectors. (**C**) Depiction of a prePAAR-containing effector being exported by the T6SS into recipient cells. Inset shows the hydrophobic TMDs of a class II prePAAR effector disrupting the inner membrane of the target bacterium to allow entry of the effector toxin domain into the cytoplasm.

recruitment of effector proteins among hundreds of cytoplasmic proteins. Here, we use a combination of genetic, biochemical and structural approaches to investigate the mechanism of recruitment for a widespread family of membrane protein effectors exported by the T6SS. Our work demonstrates that the N-terminal region of these effectors possesses two structural elements that are critical for their delivery between bacterial cells by the T6SS apparatus. First, this region contains TMDs, which interact with the Eag family of chaperones and are proposed to play a role in effector translocation across the inner membrane of recipient cells (*Quentin et al., 2018*). Additionally, this region possesses prePAAR, which we show is required for the proper functioning of PAAR, thereby facilitating the interaction of this domain with its cognate VgrG protein and enabling effector export by the T6SS.

The prePAAR motif is present in Eag-binding effectors prePAAR effectors constitute a new family of T6SS effectors that are defined by the existence of a prePAAR motif, N-terminal TMDs, a PAAR domain and a C-terminal toxin domain. Most notably, we show that this group of effectors co-occurs with Eag chaperones and that chaperone interaction with prePAAR effector TMDs is a conserved property of this protein family. While previous work has relied on genetic context to identify the cognate effector of an Eag chaperone (*Whitney et al., 2015*; *Alcoforado Diniz and Coulthurst, 2015*), our use of the prePAAR motif as an effector discovery tool enables the identification of these effectors in any genetic context. Other families of chaperones, such as the DUF4123 or DUF2169 protein families, have also been shown to affect the stability and/or export of their cognate effectors

(*Burkinshaw et al., 2018*; *Bondage et al., 2016*; *Pei et al., 2020*). However, little is known about the specificity of these chaperones for their effector targets, which do not contain predicted N-terminal TMDs. DUF4123 chaperones are encoded next to effectors with diverse domain architectures and studies on several members of this family have shown chaperone interactions occur with domains of effectors possessing no apparent shared sequence properties (*Liang et al., 2015*). A lack of structural information for these and DUF2169 chaperones has hindered an understanding of why certain T6SS effectors require members of these chaperone families for export from the cell.

## Role of Eag chaperones in binding effector TMDs and prePAAR

As demonstrated in *Figure 2*, EagR1 and EagT2 from *P. protegens* interact with and stabilize RhsA and Tne2, respectively, but not vice versa. The inability of these chaperones to interact with non-cognate effectors can be explained by the two starkly contrasting binding modes observed in our co-crystal structures. The class I prePAAR effector and RhsA homolog Rhs1 interacts with its cognate chaperone SciW in an asymmetric manner, whereas the class II effector and Tne2 homolog Tse6 adopts a pseudosymmetric binding mode whereby two separate alpha helices interact with each EagT6 chaperone protomer in a similar location. Our structural analyses suggest that Rhs1 residues F20 and F43 play a critical role in its asymmetric binding mode because the aromatic side chains of these amino acids insert into hydrophobic pockets present in SciW that are lacking in EagT6. The backs of these hydrophobic binding pockets in SciW are formed by residue G108, whereas in EagT6 the equivalent space is occupied by the side chain of F104. As such, the F20 and F43 side chains of Rhs1's TMD are able to insert into the hydrophobic pockets of SciW but are unable to do so with EagT6 due to a steric clash that would occur with F104. Conversely, the equivalent residue to F20 in the TMD of Tse6 is A26. The small side chain of alanine lacks the volume needed to fill the hydrophobic binding pocket of SciW and thus this interaction would likely contribute less free energy of binding with SciW. We expect that these same structural features contribute to the effector specificity observed for EagR1 and EagT2 as these chaperones also possess an alanine and phenylalanine at the corresponding positions in their hydrophobic pockets. Furthermore, the equivalent positions to Rhs1 in the TMD of RhsA has large hydrophobic residues (I23/L46), whereas the TMD of Tne2 has an alanine residue (A26) that aligns with A26 of Tse6. Although our structural models and bioinformatic analyses of chaperone-effector pairs suggest this pattern of Eag chaperone specificity is widespread, additional biochemical experimentation will be needed to demonstrate the generalizability of these findings.

Another important difference observed in the class I and class II chaperone-effector interactions is the position of the prePAAR region in each chaperone-effector pair. The asymmetric chaperone-TMD interaction allows SciW to 'shield' the hydrophobic regions of Rhs1's N-terminus from the aqueous milieu while also positioning its prePAAR motif in such a way that would allow it to interact with PAAR. By contrast, the pseudosymmetric binding mode of Tse6 to EagT6 appears to be much more dynamic as interpretable electron density for bound Tse6 was only observed when the effector fragment was held in place by interactions with an adjacent complex in the crystallographic asymmetric unit. Consequently, we speculate that even though the Tse6's prePAAR motif appears less accessible than that of Rhs1, it is likely highly dynamic in solution and thus may adopt a markedly different conformation when in complex with PAAR.

Despite containing a primarily beta-sheet secondary structure, Eag chaperones interact with effector TMDs by mimicking the interactions that occur between the helices of alpha-helical membrane proteins, which, to our knowledge, is a unique mechanism for a chaperone-effector interaction. Upon binding their cognate effector, we hypothesize that Eag chaperones not only shield effector TMDs from solvent but also distort their structure to prevent potential hairpin formation and erroneous insertion into the inner membrane of the effector-producing cell. Because Eag-interacting TMDs have likely evolved to insert into bacterial membranes, a mechanism to prevent self-insertion is probably necessary prior to export. Recent work studying the secretion of TMD-containing effectors of the bacterial type III and type IV secretion systems found that shielding TMDs to prevent inner membrane insertion is a critical step for proper targeting to the secretion apparatus (*Krampen et al., 2018*). However, membrane protein effectors of these secretion systems have evolved to target eukaryotic, not bacterial, membranes and thus may not require stringent control of TMD conformation prior to export. Indeed, unlike the Eag chaperones presented here, a previously

studied T3SS chaperone was shown not to distort the conformation of effector TMDs, whose conformation remained similar before and after membrane insertion (*Nguyen et al., 2015*).

Current evidence also suggests that Eag chaperones are not secreted by the T6SS (*Cianfanelli et al., 2016a*; *Quentin et al., 2018*). This leads to two important questions: (1) when do Eag chaperones dissociate from their cognate effector? (2) How do effector TMDs remain stable after their dissociation from the chaperone? Although no definitive answers exist for either of these questions, given that effector-chaperone interactions are maintained after effector-VgrG complex formation, chaperone dissociation presumably occurs immediately before or during a T6SS firing event. One way this could be accomplished is through chaperone interactions with components of the T6SS membrane and/or baseplate subcomplexes, which might induce chaperone-effector dissociation. The lumen of the T6SS apparatus may also serve to mitigate the susceptibility to degradation observed for prePAAR effectors in the absence of Eag chaperones because the inner chamber of the T6SS apparatus may shield effectors from the protein homeostasis machinery of the cell.

## prePAAR-containing proteins contain C-terminal toxin domains that act in the cytoplasm

Studies conducted in several different bacteria suggest that many T6SSs export multiple effectors during a single firing event (*Cianfanelli et al., 2016a*; *Silverman et al., 2013*; *Hood et al., 2010*). The precise subcellular location for effector delivery in recipient cells is not well understood; however, it is noteworthy that many effectors that interact with Hcp or C-terminal extensions of VgrG target periplasmic structures such as peptidoglycan or membranes (*Flaugnatti et al., 2016*; *Silverman et al., 2013*; *Brooks et al., 2013*; *LaCourse et al., 2018*). In contrast, all characterized prePAAR proteins act on cytoplasmic targets by mechanisms that include the hydrolysis of $NAD^+$ and $NADP^+$, ADP-ribosylation of FtsZ, pyrophosphorylation of ADP and ATP, and deamination of cytidine bases in double-stranded DNA (*Whitney et al., 2015*; *Ting et al., 2018*; *Ahmad et al., 2019*; *Mok et al., 2020*). This observation supports the proposal that the TMDs in prePAAR effectors function to promote toxin entry into the cytoplasm of target cells (*Quentin et al., 2018*). Two possibilities for how this occurs include a discrete toxin translocation event that takes place after the initial delivery of effectors into the target cell periplasm or that effectors are delivered directly into the target cell cytoplasm during a T6SS firing event. The large size of Rhs repeat-containing class I prePAAR effectors favors the latter model because it is unlikely that the 2–3 N-terminal TM helices found in these proteins could form a translocation pore for the C-terminal toxin domain. Instead, we propose that the TMDs of prePAAR effectors acts as molecular grease that coats the tip of the VgrG spike allowing it to effectively penetrate target cell membranes during a T6SS firing event. It should be noted that PAAR effectors with nuclease activity that lack N-terminal TMDs have been identified, suggesting that other cell entry mechanisms likely exist, and future work may address whether these proteins have important motifs or domains that permit an alternative translocation mechanism into recipient cells (*Pissaridou et al., 2018*; *Jana et al., 2019*).

## prePAAR is required for PAAR function and effector export by the T6SS

Crystal structures of single domain PAAR proteins suggest that this domain folds independently and is highly modular (*Shneider et al., 2013*). Indeed in many instances, PAAR domains appear in isolation (orphan PAAR) and do not require additional binding partners to interact with VgrG (*Wood et al., 2019*). The initial characterization of PAAR domains established seven groups of PAAR proteins, with the most abundant being orphan PAARs (55% of 1353 PAAR proteins) while the remaining groups represent PAAR proteins with N- and/or C-terminal extensions (*Shneider et al., 2013*). Our data demonstrate that PAAR domains with N-terminal extensions possess prePAAR, which we predict may be required for the proper folding of the downstream PAAR domain. Based on our structural modelling and sequence alignments, the ability of prePAAR to assist with PAAR domain folding may in part be due to its participation in coordinating the zinc ion found near the tip of this cone-shaped protein. Our sequence analysis also suggests that while orphan PAARs contain four zinc-coordinating histidine and/or cysteine residues, the PAAR domain of prePAAR effectors contains only three, suggesting that the fourth ligand required for tetrahedrally coordinated $Zn^{2+}$ is provided by prePAAR. In this way, the PAAR domain of prePAAR effectors is split into two

components, which likely come together to form a structure that can interact with VgrG and undergo T6SS-mediated export. One consequence of this 'split PAAR' domain arrangement is that the TMDs are tethered to PAAR via their N- and C-terminus, which would restrict the mobility of the TMDs and ensure their positioning on the surface of PAAR. We speculate that the proper arrangement of prePAAR effector TMDs on the surface of PAAR is likely critical for the ability of the T6SS spike complex to effectively penetrate target cell membranes during a T6SS firing event. Future studies focused on capturing high-resolution structural snapshots of assembled prePAAR-TMD-PAAR complexes will be needed to further support this proposed mechanism.

## Conclusions

In summary, our mechanistic dissection of prePAAR effectors and their cognate chaperones has revealed fundamental new insights into bacterial toxin export and membrane protein trafficking. The unique ability of T6SSs to potently target a wide range of bacteria in a contact-dependent manner may permit their use in different biomedical applications, such as the selective depletion of specific bacterial species in complex microbial communities (*Ting et al., 2020*). An in-depth understanding of the mechanisms that that underlie T6SS effector recruitment and delivery will be of critical importance for such future bioengineering efforts.

# Materials and methods

## Bacterial strains and growth conditions

*Pseudomonas* strains used in this study were derived from *P. aeruginosa* PAO1 and *P. protegens* Pf-5 (*Supplementary file 4*). Both organisms were grown in LB medium (10 g L$^{-1}$ NaCl, 10 g L$^{-1}$ tryptone, and 5 g L$^{-1}$ yeast extract) at 37°C (*P. aeruginosa*) or 30°C (*P. protegens*). Solid media contained 1.5% or 3% agar. Media were supplemented with gentamicin (30 μg mL$^{-1}$) and IPTG (250 μM) as needed.

*Escherichia coli* strains XL-1 Blue, SM10 and BL21 (DE3) Gold or CodonPlus were used for plasmid maintenance and toxicity experiments, conjugative transfer and protein overexpression, respectively (*Supplementary file 4*). All *E. coli* strains were grown at 37°C in LB medium. Where appropriate, media were supplemented with 150 μg mL$^{-1}$ carbenicillin, 50 μg mL$^{-1}$ kanamycin, 200 μg mL$^{-1}$ trimethoprim, 15 μg mL$^{-1}$ gentamicin, 0.25–1.0 mM isopropyl β-D-1-thiogalactopyranoside (IPTG), 0.1% (w/v) rhamnose or 40 μg mL$^{-1}$ X-gal.

## Bioinformatics analyses of prePAAR-containing sequences

### Phylogenetic distribution of prePAAR effector sequences retrieved from the UniprotKB database

For the analysis of all effectors in UniprotKB we used six iterations of *jackhmmer* (HmmerWeb v2.41.1) using the first 60 amino acids of Tse6 (PA0093) protein to obtain 2378 sequences. We removed any UniProtKB deprecated sequences entries (324/2378, remaining: 2,054) and further filter, cluster, and analyze the remaining 975 effector sequences as stated below (same as analysis in *Figure 1E*). In our PAAR motif search, using our first to fourth PAAR motif HMMs (see analysis below), we identified 734/975, 200/241, 30/41, and 8/11 sequences to have PAAR motifs, respectively. The remaining three sequences that did not have PAAR motifs were determined to either directly associated with a PAAR domain downstream. There were seven sequences that did not have any predicted TM. All scripts and intermediate files can be found in https://github.com/kar-atsang/effector_chaperone_T6SS/tree/master/UniProtKB (effector_chaperone_T6SS, version 1.0, *Tsang, 2020*; copy archived at https://doi.org/10.5281/zenodo.4382635).

### Phylogenetic distribution of prePAAR effector sequences retrieved from nine T6SS-containing genera

The genome assemblies of *Pseudomonas*, *Burkholderia*, *Enterobacter*, *Escherichia*, *Salmonella*, *Serratia*, *Shigella*, *Vibrio*, and *Yersinia* were downloaded from NCBI using ncbi-genome-download (https://github.com/kblin/ncbi-genome-download (ncbi_genome_download, *Blin, 2020*, v0.210)). Protein coding genes were predicted using Prodigal (v2.6.3) and the '-e 1' option (*Hyatt et al.,*

*2010*). We developed a Hidden Markov Model (HMM) for detecting effectors by using the first 61 amino acids of Tse6 (PA0093) protein and six iterations of *jackhmmer* (HmmerWeb v2.41.1). *hmmsearch* (v3.1b2) and the effector HMM were used to identify the effectors in all genome assemblies using the ' -Z 45638612 -E 1000' options and we further filtered for a bitscore greater than 40. We further filtered to include effectors that included the prePAAR (AARxxDxxxH) motif, which we searched for using regular expressions, identifying 6129 prePAAR-containing sequences across 5584 genomes. To be included in the analysis of *Figure 1D*, each genome with at least one effector had to also encode for an Eag chaperone which we searched for using Pfam's established DcrB HMM (http://pfam.xfam.org/family/PF08786#tabview=tab6) and hmmsearch with the same parameter and bitscore cutoff as the effector search. For *Figure 1E*, to reduce spurious effector predictions, we removed sequences with less than 100 amino acids. To reduce redundancy, we removed any sequences that were 100% identical and clustered sequences with 95% sequence similarity that were less than 50 amino acids different in length using CD-HIT (v4.8.1 with ' -c 0.95 -n 5 -S 50'), leaving 1166 sequences for further analysis (*Li and Godzik, 2006*). The numbers of sequences before and after filtering for the UniprotKB and sequences isolated from the eight genera listed above are indicated in *Supplementary file 3*. We identified the presence of a PAAR domain through a repetitive process of generating a PAAR motif HMMs and using *hmmsearch* (as described above) to capture the known diversity of the PAAR motif. We started broadly by using Pfam's PAAR motif HMM (http://pfam.xfam.org/family/PF05488#tabview=tab4) to identify 895/1166 PAAR motif containing sequences. For the 271 sequences that were predicted to not have a PAAR motif, we then generated an HMM using three iterations *jackhmmer* and the PAAR motif of the Tse6 (PA0093) protein (L75 to G162) to identify 219/271 PAAR motifs. We generated a third PAAR motif HMM using 60–160 amino acids of a randomly selected sequence (GCF_001214785.1 in contig NZ_CTBP01000066.1) and two iterations of *jackhmmer* that was not identified to have a PAAR motif in the previous search but was identified to have a PAAR domain using phmmer (HmmerWeb version 2.41.1). We identified 42/52 sequences had a PAAR domain using the third PAAR motif. For the fourth PAAR domain HMM, we used the 60–160 amino acid sequence of GCF_005396085.1 in the NZ_BGGV01000116 contig and three iterations of *jackhammer* to identify 8/10 sequences that had a PAAR motif. The remaining two sequences with no PAAR domain were manually analyzed and were determined to either be directly associated with a PAAR domain downstream (GCF_001425105.1) or directly beside T6SS machinery gene (GCF_001034685.1). We predicted the transmembrane (TM) helices in proteins first using TMHMM (v2.0), Phobius web server, and TMbase (https://embnet.vital-it.ch/software/) (*Krogh et al., 2001*; *Käll et al., 2007*). Using TMHMM, we defined a TM region to include TM helices that were less than or equal to 25 amino acids apart. Therefore, any TM helix that was greater than 25 amino acids apart from another TM helix would be considered part of a new TM region. Any effector considered to have no TM or three TM regions were analyzed with Phobius with the same criteria as with TMHMM. Any effector considered to have no TM or three TM regions using Phobius, were analyzed with TMbase where we used the strongly preferred model and only interpreted TM helices with a score greater than 1450. In this model, any TM helices within the first 120 amino acids is one TM region and any number of TM helices between 200 and 300 amino acids were another region. MAFFT (v7.455) was used to align the sequences using the '–auto' option and the alignment was then trimmed to remove gaps using trimAl (v1.4) and the '-gt 0.8 -cons 80' options (*Katoh and Standley, 2013*; *Capella-Gutiérrez et al., 2009*). We constructed the maximum-likelihood phylogenetic tree using FastTree (v2.1.10) and the '-gamma' option (*Price et al., 2010*). The phylogenetic tree was visualized using ggtree (*Yu, 2020*). For *Figure 1—figure supplement 1B*, we identified neighboring (within 300 base pairs) chaperone sequences for the effectors in *Figure 1E*. We removed any effectors that did not have a chaperone and we categorized the chaperones with its corresponding effectors TM prediction. Sequence logos in *Figure 1C and F* were created by using logomaker (v0.8) (*Tareen and Kinney, 2020*). All scripts and intermediate files can be found at https://github.com/karatsang/effector_chaperone_T6SS/tree/master/NCBI_8_Genera (effector_chaperone_T6SS, version 1.0, *Tsang, 2020*; copy archived at https://doi.org/10.5281/zenodo.4382635).

## Identification of potential Zn$^{2+}$-binding residues and distribution of PAAR and split PAAR sequences

To collect orphan PAAR sequences, we used the Pfam database's information on the PAAR motif (PF05488, http://pfam.xfam.org/family/PF05488#tabview=tab1) and only obtained the 1923 sequences with one PAAR motif architecture. We then aligned and trimmed the alignment of the 1923 orphan PAAR sequences. We then used the previously mentioned 2054 effector sequences from Uni-ProtKB and filtered to only use 1765 sequences with an AARxxDxxxH motif. To identify Zn$^{2+}$-binding residues in orphan and prePAAR effector sequence logos, we used logo maker (v0.8) to create sequence logos for the first 200 amino acids (*Tareen and Kinney, 2020*). To explore the relationship between the orphan PAAR sequences and the PAAR domain sequences of the prePAAR effector, we generated two phylogenetic trees. First, we truncated the prePAAR effector sequences to only the first 300 amino acids as an estimation of the PAAR domain location. Using the orphan PAAR sequences and the shortened prePAAR effector sequences, we created a phylogenetic tree (*Figure 6—figure supplement 2*) using previously described methods. Next, we wanted to determine the relationship between orphan PAAR sequences and PAAR domain sequences in the prePAAR effectors. We built a hidden Markov model using *hmmbuild* (v3.1b2) and eight PAAR domains annotated using Phyre2 (*Kelley et al., 2015*). We then searched for a PAAR domain using *hmmsearch* (v3.1b2) in the first 300 amino acids of the prePAAR effector sequences. For each sequence, we identified an envelope boundary (amino acid coordinates) of where the PAAR domain is predicted to be and we truncated the sequence at these positions. All scripts and intermediate files can be found in: https://github.com/karatsang/effector_chaperone_T6SS/tree/master/ZnBindingResidues (effector_chaperone_T6SS, version 1.0, Tsang KK, 2020; copy archived at https://doi.org/10.5281/zenodo.4382635).

## DNA manipulation and plasmid construction

Primers were synthesized and purified by Integrated DNA Technologies (IDT). Phusion polymerase, restriction enzymes and T4 DNA ligase were obtained from New England Biolabs (NEB). Sanger sequencing was performed by Genewiz Incorporated.

Plasmids used for heterologous expression were pETDuet-1, pET29b, and pSCrhaB2-CV. Mutant constructs were made using splicing by overlap-extension PCR and standard restriction enzyme-based cloning procedures were subsequently used to ligate PCR products into the plasmid of interest.

In-frame chromosomal deletion mutants in *P. aeruginosa* and *P. protegens* were made using the pEXG2 plasmid as described previously (*Hmelo et al., 2015*). Briefly, 500–600 bp upstream and downstream of target gene were amplified by standard PCR and spliced together by overlap-extension PCR. The resulting DNA fragment was ligated into the pEXG2 allelic exchange vector using standard cloning procedures (*Supplementary file 5*). Deletion constructs were transformed into *E. coli* SM10 and subsequently introduced into *P. aeruginosa* or *P. protegens* via conjugal transfer. Merodiploids were directly plated on LB (lacking NaCl) containing 5% (w/v) sucrose for *sacB*-based counter-selection. Deletions were confirmed by colony PCR in strains that were resistant to sucrose, but sensitive to gentamicin. Chromosomal point mutations or tags were constructed similarly with the constructs harboring the mutation or tag cloned into pEXG2. Sucrose-resistant and gentamicin-sensitive colonies were confirmed to have the mutations of interest by Sanger sequencing of appropriate PCR amplicons.

## Bacterial toxicity experiments

We previously showed that a D1404A mutation was sufficient to attenuate, but not abolish, the toxicity of RhsA and allows for the cloning of this toxin in the absence of its immunity gene (*Tang et al., 2018*). Therefore, to assess the toxicity of the full-length effector and a truncated variant, we cloned RhsA$^{D1404A}$ or RhsA$^{D1404A}_{\Delta NT}$ into the rhamnose-inducible pSCrhaB2-CV vector. These plasmids were co-transformed with an IPTG-inducible pPSV39 vector containing or lacking EagR1, respectively (see *Supplementary file 5*). Stationary-phase overnight cultures containing these plasmids were serially diluted 10$^{-6}$ in 10-fold increments and each dilution was spotted onto LB agar plates containing 0.1% (w/v) L-rhamnose, 250 µM IPTG, trimethoprim 250 µg mL$^{-1}$ and 15 µg mL$^{-1}$ gentamicin. Photographs were taken after overnight growth at 37°C.

## Cell fraction preparation and secretion assays

Stationary-phase overnight cultures of *E. coli* (DE3) BL21 CodonPlus, *P. aeruginosa* Δ*retS* or *P. protegens* were inoculated into 2 mL or 50 mL LB at a ratio of 1:500, respectively. Cultures were grown at 37°C (*E. coli* and *P. aerugionsa*) or 30°C (*P. protegens*) to OD 0.6–0.8. Upon reaching the desired OD, all samples were centrifuged at 7600 x *g* for 3 min. The secreted fraction in *P. aeruginosa* or *P. protegens* samples was prepared by isolating the supernatant and treating it with TCA (final conc: 10% (v/v)) as described previously (Quentin et al., 2018). The cell pellet was resuspended in 60 μL PBS, treated with 4X laemmli SDS loading dye and subjected to boiling to denature and lyse cells. For experiments examining the stability of Tse6-VgrG1a complexes, *P. aeruginosa* cells were resuspended in 60 μL PBS and subjected to six freeze-thaw cycles prior to mixing with 2X laemmli SDS loading dye. For preparation of *P. protegens* and *E. coli* cell fractions containing his-tagged complexes, cells were resuspended in lysis buffer containing 50 mM Tris-HCl (pH 8.0), 250 mM NaCl, 10 mM imidazole and purified according to the protocol described below (see *Protein expression and purification*).

## Competition assays

A tetracycline-resistant, *lacZ*-expression cassette was inserted into a neutral phage attachment site (*attB*) of recipient *P. aeruginosa* and *P. protegens* strains to differentiate these strains from unlabeled donors. *P. protegens* recipient strains also contain a Δ*pppA* mutation to stimulate T6SS effector secretion to induce a T6SS 'counterattack' from *P. protegens* donor strains (Basler et al., 2013).

For intraspecific competitions between *P. aeruginosa* or *P. protegens* donors against isogenic recipient that lack the indicated effector-immunity pairs, stationary-phase overnight cultures were mixed in a 1:1 (v/v) ratio.

Initial ratios of donors:recipients were counted by plating part of the competition mixtures on LB agar containing 40 μg mL$^{-1}$ X-gal. The remainder of each competition mixture was spotted (10 μL per spot) in triplicate on a 0.45 μm nitrocellulose membrane overlaid on a 3% LB agar plate and incubated face up at 37°C for 20–24 hr. Competitions were then harvested by resuspending cells in LB and counting colony forming units by plating on LB agar containing 40 μg mL$^{-1}$ X-gal. The final ratio of donor:recipient colony forming units were normalized to the initial ratios of donor and recipient strains.

## Protein expression and purification

All plasmids used for co-purification experiments (chaperone-effector pairs, tagged variants of *P. protegens* proteins and Tse6 prePAAR mutants), RhsA-RhsI-EagR1-VgrG complex for negative-stain EM, Hcp (PFL_6089) and RhsA$_{ΔNT}$ used for antibody development or the SciW, EagT6-Tse6$_{NT}$ complex and the SciW-Rhs1$_{NT}$ complex used for crystallization were expressed in *E. coli* BL21 (DE3) Gold or CodonPlus cells. Important differences in expression strategy used are indicated below.

## Co-purification experiments, preparation of negative stain EM samples, and preparation of samples for antibody development

Chaperone-effector pairs (e, effector; c, chaperone) from: *Pseudomonas aeruginosa* (e: PA0093, c: PA0094), *Salmonella* Typhimurium (e: SL1344_0286, c: SL1344_0285), *Shigella flexneri* (e: SF0266, c: SF3490), *Enterobacter cloacae* (e: ECL_01567, c: ECL_01566) and *Serratia proteamaculans* (e: Spro_3017, c: Spro_3016) were co-expressed using pET29b containing the predicted chaperone and pETDuet-1 harboring the full-length effector and its predicted immunity determinant. A similar co-expression strategy was employed for the RhsA$_{ΔNT}$-RhsI complex, RhsA-RhsI-EagR1-VgrG1 complex, Tse6 and the Tse6 prePAAR variants, Tsi6 and EagT6 (see *Supplementary file 5* for details). VgrG1a was expressed in isolation in pETDuet-1 and Hcp (PFL_6089) was expressed in pET29b. For *P. protegens*, all purified proteins were expressed from their native locus.

For the expression of chaperone-effector pairs and the Tse6 prePAAR mutants, a 1 mL overnight culture of expression strains was diluted in 50 mL of LB broth and grown at 37°C (*E. coli*) until OD 0.6–0.8. 40 mL overnight cultures were grown for all other of expression strains and were diluted into 2 L of LB broth and grown to OD$_{600}$0.6–0.8 in a shaking incubator at 37°C. For most samples, protein expression was induced by the addition of 1 mM IPTG and cells were further incubated for 4.5 hr at 37°C. Expression of large protein complexes (>150 kDa) in *E. coli*, such as the chaperone-

effector pairs from *Salmonella* and *Enterobacter,* RhsA$_{\Delta NT}$-RhsI and RhsA-RhsI-EagR1-VgrG1 complexes were induced at 18°C and incubated at this temperature overnight. One millilitre overnight cultures of *P. protegens* strains expressing the desired tagged protein was diluted in 50 mL of LB broth and grown at 30°C (*P. protegens*) unitl OD 0.8. Cells were harvested by centrifugation at 9800 *g* for 10 min following incubation. For the RhsA-EagR1-VgrG1 complex and the experiments containing Tse6 prePAAR mutants, the pellets for cells expressing the cognate VgrG were combined with the pellets containing effectors, as described above. Pellets from 50 mL culture were resuspended in 3.5 mL lysis buffer (50 mM Tris-HCl pH 8.0, 300 mM NaCl, 10 mM imidazole), whereas those from 2 L of culture were resuspended in 25 mL of lysis buffer prior to rupture by sonication (6 × 30 s pulses, amplitude 30%). Cell lysates were cleared by centrifugation at 39,000 *g* for 60 min and the soluble fraction was loaded onto a gravity flow Ni-NTA column that had been equilibrated in lysis buffer. Ni-NTA-bound complexes were washed twice with 25 mL of lysis buffer followed by elution in 10 mL of lysis buffer containing 400 mM imidazole. The Ni-NTA purified complex was further purified by gel filtration using a HiLoad 16/600 Superdex 200 column equilibrated in 20 mM Tris-HCl pH 8.0 150 mM NaCl or phosphate buffered saline (for samples used for antibody development only).

Preparation of samples for crystallization *sciW* (SL1344_0285) was synthesized with codon optimization for *E. coli* and cloned into the vector pRSETA with the restriction sites NdeI/HindIII (Life Technologies). This construct includes an N-terminal 6-his tag and an HRV 3C protease cleavage site (MGSSHHHHHHSSDLEVLFQGPLS). SciW-Rhs1$_{NT}$ and EagT6-Tse6$_{NT}$ complexes were co-expressed using pETDUET-1. Note that the EagT6 construct has a C-terminal VSV-G tag (see *Supplementary file 5*). Cells were grown in LB broth to OD$_{600}$ 0.6 at 37°C at which point protein expression was induced by the addition of 1 mM IPTG. The temperature was reduced to 20°C and cultures were allowed to grow overnight. Cells were harvested by centrifugation and resuspended in lysis buffer followed by lysis with an Emulsiflex-C3 (Avestin). The lysate was cleared by centrifugation at 16,000 rpm for 30 min and the supernatant passed over a nickel NTA gravity column (Goldbio) followed by washing with 50 column volumes of chilled lysis including PMSF, DNase I, and MgCl$_2$. Proteins were eluted with five column volumes elution buffer then purified by gel-filtration using an SD75 16/60 Superdex gel filtration column equilibrated in gel-filtration buffer (GF) with an AKTA pure (GE Healthcare). For SciW, after affinity purification the protein was dialyzed in GF buffer O/N at 4°C and the His-tag removed during dialysis using HRV 3C protease. The digested SciW was passed over a nickel NTA gravity column and the flow through was collected. SciW was further purified using an SD75 16/60 Superdex gel filtration column equilibrated in GF buffer.

The buffers used were as follows: SciW lysis buffer (20 mM Tris pH 7.5, 500 mM NaCl, 20 mM imidazole); SciW elution buffer (20 mM Tris pH 7.5, 500 mM NaCl, 500 mM imidazole); SciW GF buffer (20 mM Tris pH 7.5, 250 mM NaCl, 1 mM 2-Mercaptoethanol); SciW-Rhs1$_{NT}$ and EagT6-Tse6$_{NT}$ complexes lysis buffer (20 mM Tris pH 8.0, 150 mM, 25 mM imidazole); elution buffer (20 mM Tris pH 8.0, 150 mM, 500 mM imidazole); and GF buffer (20 mM Tris pH 8.0, 150 mM NaCl, 1 mM 2-Mercaptoethanol).

## Crystallization and structure determination

SciW was concentrated to 7, 14, and 22 mg mL$^{-1}$ for initial screening using commercially available screens (Qiagen) by sitting-drop vapor diffusion using a Crystal Gryphon robot (Art Robbins Instruments). The crystallization conditions for SciW were 22 mg mL$^{-1}$ with a 1:1 mixture of 0.1 M Tris HCL pH 8.5, 25% (v/v) PEG 550 MME at 4°C. EagT6-Tse6$_{NT}$ complex was concentrated to 5, 10 and 20 mg mL$^{-1}$ and screened for crystallization conditions as per SciW. The final crystallization conditions were 20 mg mL$^{-1}$ with a 1:1 mixture of 0.2M Magnesium chloride, 0.1M Bis-Tris pH 5.5, and 25% (w/v) PEG 3350 at 4°C. SciW-Rhs1$_{NT}$ complex was concentrated to 15, 20, and 25 mg mL$^{-1}$ and screened for crystallization as per SciW. The crystallization conditions were 25 mg mL$^{-1}$ protein with a 1:1 mixture of 0.2M Ammonium sulfate, 0.1M Bis-Tris pH 5.5, and 25% (w/v) PEG 3350 at 4°C.

Diffraction data from crystals of SciW and EagT6-Tse6$_{NT}$ complex were collected in-house at 93K using a MicroMax-007 HF X-ray source and R-axis 4++ detector (Rigaku). Diffraction data from SciW-Rhs1$_{NT}$ crystals were collected at the Canadian Light Source at the Canadian Macromolecular Crystallography Facility Beam line CMCF-ID (08ID-1). SciW crystals were prepared by cryo-protection in mother liquor plus 38% PEG 550 MME and flash freezing in liquid nitrogen. Crystals of EagT6-Tse6$_{NT}$ and SciW-Rhs1$_{NT}$ complexes were prepared in the same manner with increasing the concentration of PEG3350 to 35–38%. All diffraction data were processed using XDS (*Kabsch, 2010*). Phases for

SciW were determined by the molecular replacement-single anomalous diffraction (MR-SAD) technique. A home-source data set was collected from SciW crystals soaked in cryo-protectant containing 350 mM NaI for one-minute before flash freezing. EagT6 (PDB: 1TU1) was used as a search model and phases were improved by SAD using the Phenix package (*Adams et al., 2010*). Phases for both the EagT6-Tse6$_{NT}$ and SciW-Rhs1$_{NT}$ complexes were obtained by molecular replacement using EagT6 (PDB: 1TU1) and SciW as search models, respectively, with the Phenix package. Initial models were built and refined using Coot, Refmac and the CCP4 suite of programs, Phenix, and TLS refinement (*Emsley et al., 2010*; *Murshudov et al., 1997*; *Winn et al., 2011*; *Winn et al., 2001*). Data statistics and PDB codes are listed in *Table 1*. The ligands identified included polyethylene glycol (PEG) for SciW and sulfate ions for the SciW-Rhs$_{NT}$ complex. Additionally, the side-chain of residue C11 in SciW chain B was observed to be covalently bound to 2-Mercaptoethanol to form CME (S,S-(2-hydroxyethyl)thiocysteine). The coordinates and structure factors have been deposited in the Protein data Bank, Research Collaboratory for Structural Bioinformatics, Rutgers University, New Brunswick, NY (http://www.rcsb.org). Molecular graphics and analysis were performed using Pymol (Schrödinger, LLC) and UCSF Chimera (*Pettersen et al., 2004*).

## Electron microscopy and image analysis
### Negative stain sample preparation
Four microlitres of each protein sample at a concentration of approximately 0.01 mg mL$^{-1}$ was applied onto glow-discharged carbon-coated copper grids. After 45 s of incubation at room temperature, excess liquid was blotted away using Whatman No. four filter paper, followed by two washing steps with GF buffer. Samples were then stained with 1% (w/v) uranyl formate solution and grids stored at RT until usage.

### Data collection and image analysis
Images were recorded manually with a JEOL JEM-1400 microscope, equipped with a LaB$_6$ cathode and 4k × 4 k CMOS detector F416 (TVIPS), operating at 120 kV. For VgrG1, RhsA$_{\Delta NT}$, the EagR1-RhsA complex and EagR1-RhsA-VgrG1 complex, a total of 99, 140, 100 and 120 micrographs, respectively, were collected with a pixel size of 2.26 Å. Particles for the VgrG1, RhsA$_{\Delta NT}$, EagR1-RhsA complex and EagR1-RhsA-VgrG1 complex were selected automatically with crYOLO using individually pre-trained models, resulting in 18,676, 23,907, 32,078, and 31,409 particles, respectively (*Wagner et al., 2019*). Subsequent image processing was performed with the SPHIRE software package (*Moriya et al., 2017*). Particles were then windowed to a final box size of 240 × 240 pixel. Reference-free 2-D classification was calculated using the iterative stable alignment and clustering algorithm (ISAC) implemented in SPHIRE, resulting in 2-D class averages of each respective complex (*Yang et al., 2012*). Distance measurement were performed with the e2display functionality in EMAN2 (*Tang et al., 2007*). The placement of the crystal structure into the electron density map (EMD-0135) was done using rigid-body fitting in Chimera (*Pettersen et al., 2004*). Here, Tse6-TMD and EagT6 of the EagT6-TMD crystal structure were fitted independently as rigid bodies to better describe the density. Due to the distinct shape of the PAAR domain, three different orientations were possible in the docking step, each rotated by 120˚. Docking of Tse6-TMD into the density embraced by the second EagT6 described this density less well.

## Western blot analyses
Western blot analyses of protein samples were performed as described previously for rabbit anti-Tse1 (diluted 1:5,000; Genscript), rabbit anti-FLAG (diluted 1:5,000; Sigma), rabbit anti-VSV-G (diluted 1:5,000; Sigma), rabbit anti-Hcp1 (*P. aeruginosa*) (diluted 1:5,000, Genscript) and detected with anti-rabbit horseradish peroxidase-conjugated secondary antibodies (diluted 1:5,000; Sigma) (*Ahmad et al., 2019*). Rabbit anti-Hcp (*P. protegens*) was used at a 1:5000 dilution. Western blots were developed using chemiluminescent substrate (Clarity Max, Bio-Rad) and imaged with the ChemiDoc Imaging System (Bio-Rad).

## Acknowledgements

The authors thank Jianhua Zhao for electron microscopy expertise, Sarah Trilesky and Matthew Walker for their assistance with cloning and protein purification and Peter Stogios, Seemay Chou, James Holton and Atanas Radkov for crystallography expertise. SA and KKT were supported by Ontario Graduate Scholarships and AGM holds a Cisco Research Chair in Bioinformatics. Part of the research described in this paper was performed using beamline 08ID-1 at the Canadian Light Source, a national research facility of the University of Saskatchewan, which is supported by the Canada Foundation for Innovation (CFI), the Natural Sciences and Engineering Research Council (NSERC), the National Research Council (NRC), the Canadian Institutes of Health Research (CIHR), the Government of Saskatchewan, and the University of Saskatchewan. This work was supported by the Max Planck Society (to SR) and by grants from CFI (34531 to AGM and 37841 to GP), NSERC (RGPIN-2017–05350 to JCW and RGPIN-2018–04968 to GP) and CIHR (PJT156129 to JCW and PJT156214 to AGM). Computer resources were supplied by the McMaster Service Lab and Repository computing cluster, funded in part by grants to AGM from CFI and Compute Canada (http://www.computecanada.ca).

## Additional information

### Funding

| Funder | Grant reference number | Author |
| --- | --- | --- |
| Canadian Institutes of Health Research | PJT-156129 | John C Whitney |
| Natural Sciences and Engineering Research Council of Canada | RGPIN-2017-05350 | John C Whitney |
| Natural Sciences and Engineering Research Council of Canada | RGPIN-2018-04968 | Gerd Prehna |
| Canadian Institutes of Health Research | PJT156214 | Andrew G McArthur |
| Max Planck Society | | Stefan Raunser |
| Canadian Foundation for Innovation | 34531 | Andrew G McArthur |
| Canadian Foundation for Innovation | 37841 | Gerd Prehna |

The funders had no role in study design, data collection and interpretation, or the decision to submit the work for publication.

### Author contributions

Shehryar Ahmad, Conceptualization, Data curation, Formal analysis, Investigation, Visualization, Methodology, Writing - original draft, Writing - review and editing; Kara K Tsang, Data curation, Formal analysis, Investigation, Visualization, Methodology, Writing - review and editing; Kartik Sachar, Tahmid M Tashin, Nathan P Bullen, Data curation, Formal analysis, Investigation; Dennis Quentin, Data curation, Formal analysis, Investigation, Writing - review and editing; Stefan Raunser, Andrew G McArthur, Supervision, Funding acquisition, Project administration, Writing - review and editing; Gerd Prehna, Conceptualization, Data curation, Formal analysis, Supervision, Funding acquisition, Investigation, Methodology, Writing - original draft, Project administration, Writing - review and editing; John C Whitney, Conceptualization, Formal analysis, Supervision, Funding acquisition, Writing - original draft, Project administration, Writing - review and editing

### Author ORCIDs

Shehryar Ahmad https://orcid.org/0000-0002-2747-5212
Kara K Tsang https://orcid.org/0000-0001-7742-8855

Kartik Sachar (iD) https://orcid.org/0000-0003-4077-1734
Dennis Quentin (iD) http://orcid.org/0000-0003-3825-7066
Stefan Raunser (iD) http://orcid.org/0000-0001-9373-3016
Gerd Prehna (iD) https://orcid.org/0000-0001-5539-7533
John C Whitney (iD) http://orcid.org/0000-0002-4517-8836

## Decision letter and Author response

Decision letter https://doi.org/10.7554/eLife.62816.sa1
Author response https://doi.org/10.7554/eLife.62816.sa2

## Additional files

### Supplementary files

• Supplementary file 1. List of prePAAR motif-containing proteins identified in the UniProtKB Database . The document contains two separate sheets. List A corresponds to 2054 prePAAR-containing sequences that were identified through an iterative search of the UniprotKB using $Tse6_{NT}$. List B corresponds to 975 sequences collected following filtering of list A (see Materials and Methods for details).

• Supplementary file 2. List of prePAAR motif-containing proteins from assembled genomes of all species belonging to the genera *Burkholderia*, *Escherichia*, *Enterobacter*, *Pseudomonas*, *Salmonella*, *Serratia, Shigella*, and *Yersinia*. The document contains two separate sheets. List C corresponds to 6101 prePAAR-containing sequences that were identified through an iterative search of the UniprotKB using $Tse6_{NT}$. List D corresponds to 1166 sequences collected following filtering of list C (see Materials and methods for details).

• Supplementary file 3. Summary of the number of genomes and effector sequences used in our informatics analyses. This document contains three separate sheets. The 'UniprotKB-effectors' sheet shows the quantity of initial prePAAR-containing sequences that were identified in our search and the number of sequences that were used following filtering and removal of redundancy (plotted in the cladogram in *Figure 1—figure supplement 1A*). The numbers in bold indicate the number of sequences in *Supplementary file 1*. The 'eight genera - genomes' sheet corresponds to the number of genomes from the eight genera (*Burkholderia*, *Escherichia*, *Enterobacter*, *Pseudomonas*, *Salmonella, Serratia, Shigella*, and *Yersinia*) that contained one prePAAR-containing sequence and the number that remained following filtering and removal of redundancy. The '8-genera – effectors' sheet corresponds to initial and final numbers of prePAAR-containing sequences that were identified in the eight genera listed above. The final number of sequences in this sheet were used to construct the cladogram in *Figure 1E*. The numbers in bold indicate the numbers of sequences in the lists in *Supplementary file 2*.

• Supplementary file 4. Strains used in this study.

• Supplementary file 5. Plasmids used in this study.

• Transparent reporting form

### Data availability

X-ray diffraction data for the SciW, SciW:Rhs1 complex, and Tse6:EagT6 complex have been deposited in the PDB under the accession codes 6XRB, 6XRR and 6XRF, respectively.

The following datasets were generated:

| Author(s) | Year | Dataset title | Dataset URL | Database and Identifier |
|---|---|---|---|---|
| Ahmad S, Tsang KK, Sachar K, Quentin D, Tashin TM, Bullen NP, Raunser S, McArthur AG, Prehna G, Whitney JC | 2020 | Crystal structure of SciW from Salmonella typhimurium | https://www.rcsb.org/structure/6XRB | RCSB Protein Data Bank, 6XRB |

| Ahmad S, Tsang KK, Sachar K, Quentin D, Tashin TM, Bullen NP, Raunser S, McArthur AG, Prehna G, Whitney JC | 2020 | Structure of SciW bound to the Rhs1 Transmembrane Domain from Salmonella typhimurium | https://www.rcsb.org/structure/6XRR | RCSB Protein Data Bank, 6XRR |
| Ahmad S, Tsang KK, Sachar K, Quentin D, Tashin TM, Bullen NP, Raunser S, McArthur AG, Prehna G, Whitney JC | 2020 | EagT6 Tse6 NT complex | https://www.rcsb.org/structure/6XRF | RCSB Protein Data Bank, 6XRF |

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
