## [Decision Letter]

Thank you for submitting your article "Structural basis for effector transmembrane domain recognition by type VI secretion system chaperones" for consideration by *eLife*. Your article has been reviewed by two peer reviewers, including Rajan Sankaranarayanan as the Reviewing Editor and Reviewer #1, and the evaluation has been overseen by John Kuriyan as the Senior Editor. The following individual involved in review of your submission has agreed to reveal their identity: Dor Salomon (Reviewer #3).

The reviewers have discussed the reviews with one another and the Reviewing Editor has drafted this decision to help you prepare a revised submission.

Summary:

The manuscript deals with the elucidation of the mechanism by which bacteria deliver a set of effectors using type VI secretion system to other bacteria. Using bioinformatics approaches, the authors have identified a prePAAR motif in a set of effectors and classified them into two major families. Using structural, biochemical and in vivo approaches, the authors show the importance of the motif in loading the effector molecule on to the type VI assembly using Eag chaperone. The results are also interesting in demonstrating that the chaperone-effector pairing is specific. Overall, the work provides an in-depth molecular understanding on previously open questions in the T6SS field, especially those related to chaperone/adapter protein activity.

Essential revisions:

1) The identification and the role of prePAAR motif is central to the manuscript. Therefore, mutational data of key residues in the motif showing its role in the direct interaction with the PAAR domain will significantly strengthen the major claim of the manuscript (see reviewer #1 concern b). However, in the present situation, if the authors feel that they will not be able to do this in a short frame of time, you can tone down the claims accordingly.

2) The schematic given in Figure 7 can be modified based on the comments of the reviewer.

3) Make appropriate revisions by addressing the other concerns of both the reviewers.

Reviewer #1:

This manuscript significantly contributes to our understanding of the mechanism by which a subset of effectors is delivered by type VI secretion system (T6SS) from one bacteria to another. The authors have identified a prePAAR motif in Eag-dependent TMD containing effectors of T6SS and segregate them into two broad classes. Further, the authors provide in vitro and in vivo evidence to show that the effector and Eag binding is specific. Earlier, the same group has shown that TMD containing effectors require chaperone assistance for loading on to VgrG protein of the type VI assembly and for eventually getting delivered to the recipient bacteria. Here, the authors have also solved the crystal structure of both class I and class II effector molecules in complex with their cognate chaperones which revealed the atomic details of the interaction. Interestingly, the newly identified prePAAR motif is shown to be essential for loading the effector molecule onto VgrG and its delivery. Overall, the work is well done and I would recommend publication after addressing the concerns that are listed below.

a) The authors justify the use of "prePAAR" by quoting the earlier work by Zhang et al., 2011. However, it looks like the term prePAAR was not introduced then and the Zhang et al. work refers to the N-terminal region to PAAR domain, which includes the newly identified motif (current study) as well as the TMD. Therefore, the authors have to clearly define what exactly do they mean by prePAAR at the first instance and follow the same throughout. To provide better clarity, since “prePAAR” indicates everything N-terminal to PAAR domain therefore it may be better to use an alternative term for the newly identified motif.

b) Based on the data shown in Figure 6F, the authors conclude that prePAAR interacts with PAAR domain. However, this claim needs to be directly proven by including a pull-down of PAAR domain with an N-terminal construct having prePAAR point mutants (as in Figure 6A, B and G) or just prePAAR, instead of the N-terminal construct (prePAAR + TMD).

c) One interesting finding of the manuscript is the fidelity of the chaperone-effector pairing. However, the authors fail to comment on this fidelity based on the crystal structure information they have obtained. By attempting a simple modelling approach of cross-pairs, they may be able to provide a rationale in terms of the structural basis for this cognate pairing.

d) The authors have an interesting proposition that H15 of prePAAR motif participates in Zn+2 binding that eventually helps in PAAR domain folding. However, it is perplexing that the D11A mutant of the effector is compromised in VgrG binding and also inducing toxicity in the target bacteria, while the H15A mutant is similar to wild type (Figure 6A, B and G). Therefore, this proposition needs a reconsideration. Can the authors provide any evidence for the importance of Zn+2 binding in the folding of the PAAR domain?

e) The authors should consider the following suggestions for preparing a more coherent schematic of the entire finding depicted in Figure 7. (1) prePAAR can be depicted at the tip of PAAR domain as the current depiction is misleading where prePAAR is shown at the interface of PAAR domain and VgrG. However, the authors do not seem to have provided any evidence for prePAAR and VgrG interaction. (2) As per the structural knowledge (from this work and previous studies), it is clear that the chaperones are symmetric dimers and therefore the schematics should represent the same throughout the manuscript. (3) From the model (Figure 7B), it appears that binding of prePAAR to PAAR domain is dependent on TMD-chaperone binding. However, there is no experimental evidence for such sequential binding. (4) Authors should provide an explanation for their statement that split PAAR is an “evolved” version of PAAR domain and the same needs to be included in the Discussion. Why it cannot be the other way around? Also, it is worth considering the possibility that in split-PAARs the stabilization of PAAR domain with prePAAR may provide structural rigidity to the TMD for effective insertion into the membrane.

f) In the Abstract, the authors state "…our findings define the mechanism of chaperone assisted secretion of a widespread family of T6SS membrane protein effectors". It appears that the significant part of this claim is already made in their earlier work (Quentin et al., 2018) and therefore the authors need to make a clear distinction.

g) The authors have shown that mutations in pre-PAAR affect the binding of the effector to VgrG. However, there is no evidence which shows that prePAAR has any direct role in PAAR domain folding and therefore the lines in the Abstract, "…we find that prePAAR functions to facilitate proper folding of the downstream PAAR domain…" needs to be looked at.

h) In the Abstract and Introduction, the author should also mention the number of genera which are spanned by these 6000 putative TMD containing effectors as this number does not directly signify the impact of the work.

Reviewer #3:

The results presented by Ahmad et al. provide an in-depth molecular understanding and novel concepts on previously open questions in the T6SS field, especially those related to chaperone/adapter protein activity. The work is appropriate for publication in *eLife*, and the authors provide a convincing body of evidence to support their claims. The work is solid, nicely presented and easy to follow. I note that I am not a structural biologist and therefore unable to provide an expert opinion on the structural portions of the work, although they seem reasonable. I have no major comments and no additional experiments are required.

---

## [Author Response]

Essential revisions:1) The identification and the role of prePAAR motif is central to the manuscript. Therefore, mutational data of key residues in the motif showing its role in the direct interaction with the PAAR domain will significantly strengthen the major claim of the manuscript (see reviewer #1 concern b). However, in the present situation, if the authors feel that they will not be able to do this in a short frame of time, you can tone down the claims accordingly.

We agree that a point mutant control is an important addition to Figure 6F that would bolster the claims presented in the manuscript. As we describe in detail in our responses to reviewer #1, we were able to partially address this reviewer’s concern by generating new chromosomal mutations in *P. aeruginosa* that show prePAAR is both necessary and sufficient for PAAR-dependent Tse6-VgrG interactions in vivo, even in the absence of effector TMDs and EagT6 (new Figure 6D). However, in contrast to our in vivo findings, we found that the Tse6^D11A^ and Tse6^D11A, H15A^ prePAAR mutants still interact with PAAR in vitro. These experiments using purified components have revealed the complexities associated with assessing interdomain effector interactions in vitro and a proper dissection of this interaction will require extensive structural and biophysical characterization that is beyond the scope of this work. In light of this, we have removed the data from Figure 6F in the original manuscript, readjusted the figure and added in a new Figure 6D that shows our new in vivo results. As per the suggestion of the reviewer, we have also toned down our claims for the role of prePAAR in catalyzing the folding of PAAR.

2) The schematic given in Figure 7 can be modified based on the comments of the reviewer.

We have adjusted the figure according the reviewer’s recommendations. As described below, the only part we did not change was the sequential binding of the chaperone to the TMD because our current data suggest that chaperone-TMD binding should precede prePAAR-PAAR binding.

3) Make appropriate revisions by addressing the other concerns of both the reviewers.

All other concerns were also addressed in the revised text.

Reviewer #1:This manuscript significantly contributes to our understanding of the mechanism by which a subset of effectors is delivered by type VI secretion system (T6SS) from one bacteria to another. The authors have identified a prePAAR motif in Eag-dependent TMD containing effectors of T6SS and segregate them into two broad classes. Further, the authors provide in vitro and in vivo evidence to show that the effector and Eag binding is specific. Earlier, the same group has shown that TMD containing effectors require chaperone assistance for loading on to VgrG protein of the type VI assembly and for eventually getting delivered to the recipient bacteria. Here, the authors have also solved the crystal structure of both class I and class II effector molecules in complex with their cognate chaperones which revealed the atomic details of the interaction. Interestingly, the newly identified prePAAR motif is shown to be essential for loading the effector molecule onto VgrG and its delivery. Overall, the work is well done and I would recommend publication after addressing the concerns that are listed below.a) The authors justify the use of "prePAAR" by quoting the earlier work by Zhang et al., 2011. However, it looks like the term prePAAR was not introduced then and the Zhang et al. work refers to the N-terminal region to PAAR domain, which includes the newly identified motif (current study) as well as the TMD. Therefore, the authors have to clearly define what exactly do they mean by prePAAR at the first instance and follow the same throughout. To provide better clarity, since “prePAAR” indicates everything N-terminal to PAAR domain therefore it may be better to use an alternative term for the newly identified motif.

The reviewer is correct, Zhang et al. use the term “PrePAARTM”, which is used to refer to both prePAAR and the first TMD region preceding PAAR. Because our work demonstrates that prePAAR and the TMDs likely have distinct roles in T6S, we decided to refer to prePAAR and the TMD regions separately. We apologize for our lack of clarity with regards to our nomenclature and we have clarified this in our revised manuscript (subsection “prePAAR is a motif found in TMD-containing effectors that interact with Eag chaperones”).

b) Based on the data shown in Figure 6F, the authors conclude that prePAAR interacts with PAAR domain. However, this claim needs to be directly proven by including a pull-down of PAAR domain with an N-terminal construct having prePAAR point mutants (as in Figure 6A, B and G) or just prePAAR, instead of the N-terminal construct (prePAAR + TMD).

The reviewer brings up a good point. Over the past month, we conducted a set of experiments designed to address this concern and these experiments revealed two important insights. First, we found that a prePAAR-PAAR fusion can form stable Tse6-VgrG complexes in vivo in *P. aeruginosa.* Second, we found that the nature of the interaction between the prePAAR motif and the PAAR domain are more nuanced that previously appreciated and challenging to assess in vitro. We have summarized three separate experiments that were attempted to address the reviewer’s point and based on the data described below, we have removed the blot in Figure 6F, readjusted the figure by moving some panels to Figure 6—figure supplement 1 for clarity and adding a new Figure 6D, which contains our new in vivo findings. A comprehensive structural and biophysical characterization of prePAAR-PAAR interactions will be the subject of future work in our lab but we feel that this is beyond the scope of the current study. As is, we believe that the in vivo data shown in Figure 6A and D and the in vitro data shown in 6E is sufficient to highlight the importance of prePAAR in affecting the PAAR-dependent Tse6-VgrG interaction and have removed parts of the text that alluded to prePAAR having a direct role in PAAR folding.

Experiment 1: we generated a *P. aeruginosa* chromosomal mutant in *tse6* that results in prePAAR being fused directly to PAAR by deleting both effector TMDs and conducted a western blot to assess the formation of a PAAR-dependent Tse6-VgrG complex in vivo. As we have shown in this work and our previous work (Quentin et al., 2018), the formation of this complex allows us to determine if PAAR is functional in vivo and in the context of this work, can be used to determine if prePAAR influences PAAR function. We found that the SDS-resistant Tse6-VgrG complex forms in a manner that is dependent on the prePAAR motif. This suggests that in conjunction with PAAR, prePAAR alone is sufficient for VgrG interaction with Tse6 and therefore it likely affects PAAR folding.

Experiment 2: we co-expressed Eag-VSV-G and PAAR-FLAG with a double mutation-containing N-terminal fragment of Tse6 (NT-Tse6 ^D11A, H15A^) and conducted a pull-down using the N-terminal Tse6 fragments as bait, as done for the wild-type fragment in Figure 6F. We found that both the wild-type and double mutant pulled-down PAAR. Based on these data, we concluded that studying these two purified protein fragments in vitro does not recapitulate the molecular interactions that we observe in cells (Figure 6A).

Experiment 3: we fused prePAAR to the C-terminus of maltose-binding protein (MBP). We used prePAAR sequences from Tse6 and Rhs1 and attempted pull-down experiments using the Tse6 or Rhs1 PAAR proteins. We used either His_6_-tagged MBP fusions or PAAR (the prey protein in either case was FLAG-tagged) for this experiment. The constructs were tested both when co-expressed or expressed separately and co-purified. We found that in all cases, the presence of the prePAAR motif did not affect the amount of PAAR or MBP-prePAAR recovered. As with experiment 2, these data also indicate that our in vitro system is not recapitulating what we observe in *P. aeruginosa*.

For experiment 1, it is important to note that we detected less Tse6-VgrG complex in a strain expressing the prePAAR-PAAR fusion compared to the parental background. This may be attributable to a miscalculation in determining the ideal boundaries to truncate the effector and suggests that the PAAR domain is sensitive to misfolding when the effector is truncated. Therefore, for experiments 2 and 3, it is possible that the boundaries we used to study the PAAR domain in isolation inadvertently exposed surfaces on this protein that promote promiscuous binding independent of the prePAAR motif. Our predictions for the PAAR domain boundaries are based on orphan PAAR sequences, which, as shown in the manuscript, differ from the PAAR sequences found in Tse6/Rhs1-like effectors. The cryo-EM density for the Tse6 PAAR domain from our previous work (Quentin et al., 2018) is also too low resolution to make an accurate prediction for domain boundaries. Further experimentation will be needed to identify the exact boundaries that produce stable variants of these domains that behave similar to the full-length proteins in vivo. We also realize we can further extend experiment 1 to include a D11A/H15A double mutation. However, we felt that the generation of this mutant in *P. aeruginosa* would be redundant with our point mutant data on the full-length effector shown in Figure 6A. Additionally, because the strain expressing the Tse6 prePAAR-PAAR fusion has significantly less detectable Tse6-VgrG complex, we would prefer to test multiple fusions of prePAAR-PAAR and subsequently generate the double mutant in those strains as well to conduct a more comprehensive analysis of prePAAR-PAAR binding both in vivo and in vitro.

Based on these experiments and the conclusions we have drawn from them, we have adjusted our manuscript in order to tone down our assertion that prePAAR plays a role in catalyzing the folding of PAAR domains. These changes were made to the Abstract, Introduction, Results and Discussion.

c) One interesting finding of the manuscript is the fidelity of the chaperone-effector pairing. However, the authors fail to comment on this fidelity based on the crystal structure information they have obtained. By attempting a simple modelling approach of cross-pairs, they may be able to provide a rationale in terms of the structural basis for this cognate pairing.

We agree with the reviewer and have added additional discussion on how our co-crystal structures may provide a model for Eag-effector cognate pairing. We have added these points in the Discussion section entitled “Role of Eag chaperones in binding effector TMDs and prePAAR” Briefly, there are symmetrical hydrophobic pockets present in chaperones that interact with class I prePAAR effectors that allow for the insertion of large hydrophobic residues present in the effector TMD (F/I indicated by black arrow in the alignment below). These hydrophobic pockets are absent in chaperones that interact with class II effectors and thus we predict that a steric clash would occur that would prevent non-cognate pairing.

We have a provided a small figure of the TMD alignments in Author response image 1.

**Author response image 1. sa2fig1:** Figure showing the alignment of Class 1 (SciW and EagR1) and Class 2 (EagT6 and Tne2) prePAAR-TMD regions studied in this manuscript. The black arrows indicate the residues that may contribute to asymmetric vs. pseudosymmetric binding.

d) The authors have an interesting proposition that H15 of prePAAR motif participates in Zn+2 binding that eventually helps in PAAR domain folding. However, it is perplexing that the D11A mutant of the effector is compromised in VgrG binding and also inducing toxicity in the target bacteria, while the H15A mutant is similar to wild type (Figure 6A, B and G). Therefore, this proposition needs a reconsideration. Can the authors provide any evidence for the importance of Zn+2 binding in the folding of the PAAR domain?

The reviewer brings up an important point about the role of Zn^2+^ binding in PAAR domain folding. Our structural model of the Tse6 PAAR domain and its structural alignment with the orphan PAAR c1882 shows that the H15 residue in Tse6 structurally overlaps with the first histidine in c1882, which is known to coordinate Zn^2+^ (shown in Figure 6—figure supplement 1B). These data, in addition to the informatics analysis showing the conservation of the H residue in prePAAR and orphan PAARs (Figure 6—figure supplement 1C), only suggest that H15 in Tse6 takes part in Zn^2+^ binding. However, we show that mutation of this residue in the prePAAR motif of Tse6 does not affect the formation of a Tse6^H15A^-VgrG complex in vivo or in vitro but does sensitize this complex to denaturation by boiling (Figure 6—figure supplement 1D and E). Thus, the PAAR domain of the Tse6^H15A^ complex may lack some structural integrity but maintains its overall architecture and its ability to bind VgrG.

In the initial characterization of PAAR proteins, it was found that Zn^2+^-binding residues are conserved across homologs and always appear to be localized to the “tip” of the PAAR protein (Shneider et al., 2013). Because there are multiple residues that bind Zn^2+^ and these residues are all located far away from the base of the protein, it could be that a single mutation at this site would not significantly alter the folding of the PAAR domain and thus abrogate its binding to VgrG. The D11 residue, however, is predicted to be localized near the base of the PAAR domain, where it potentially mediates important intramolecular interactions that are necessary to stabilize the folding of this region and mediate its interaction with VgrG. It is possible that the H15 residue tethers the prePAAR motif to the PAAR domain, whereas the D11 residue contributes important electrostatic interactions that are necessary for overall PAAR domain folding.

At this time, the question of whether Zn^2+^ affects PAAR folding is out of the scope of this paper because we feel that the high-resolution structural characterization of an Eag chaperone in complex with the prePAAR-TMD-PAAR region of an effector is the most direct way to determine the specific interactions between prePAAR residues and the PAAR domain or its bound zinc ion.

e) The authors should consider the following suggestions for preparing a more coherent schematic of the entire finding depicted in Figure 7. (1) prePAAR can be depicted at the tip of PAAR domain as the current depiction is misleading where prePAAR is shown at the interface of PAAR domain and VgrG. However, the authors do not seem to have provided any evidence for prePAAR and VgrG interaction. (2) As per the structural knowledge (from this work and previous studies), it is clear that the chaperones are symmetric dimers and therefore the schematics should represent the same throughout the manuscript. (3) From the model (Figure 7B), it appears that binding of prePAAR to PAAR domain is dependent on TMD-chaperone binding. However, there is no experimental evidence for such sequential binding. (4) Authors should provide an explanation for their statement that split PAAR is an “evolved” version of PAAR domain and the same needs to be included in the Discussion. Why it cannot be the other way around? Also, it is worth considering the possibility that in split-PAARs the stabilization of PAAR domain with prePAAR may provide structural rigidity to the TMD for effective insertion into the membrane.

The reviewer brings up several important points about the figure and we have addressed their concerns in sequence. (1) We have changed the position of prePAAR in the figure to depict an interaction solely with the PAAR domain and not VgrG. (2) We have changed the cartoon of the chaperone to be symmetric in both Figures 1B, 7B and C. (3) While the reviewer is correct in that there is no direct evidence to show that the chaperone must bind the TMD before prePAAR can bind PAAR, we believe that it is important to show this interaction in this way. The chaperone gene and effector gene are found in the same operon and co-transcribed, as shown in Figure 1A. Based on our current and previous work, we know that the TMDs of prePAAR effectors are unstable and must be stabilized by the chaperone. If the chaperone is translated first, it is likely that it will bind the TMDs as they are translated, rather than bind a fully translated prePAAR-TMD-PAAR effector. Binding prePAAR and the TMDs in this way may orient these regions to facilitate proper binding with PAAR (as suggested by Figure 5H). (4) We agree with the reviewer that an explanation of the relationship between orphan PAARs and split PAARs is required in the Results and in the Discussion. To understand the relationship between these PAAR proteins, we constructed a phylogram of orphan and split PAAR (prePAAR-TMD-PAAR) proteins. For this analysis, we used the 564 orphan PAAR and 1,765 split PAAR sequences from Figure 6—figure supplement 1C. The split PAAR sequences are significantly longer than the orphan PAAR sequences due to the presence of the TMD. To more accurately identify boundaries for the PAAR domain in the split PAAR sequences, we built a hidden Markov model using the representative PAAR proteins from our analysis in Figure 1G. We used the hidden Markov model established from this analysis to identify the PAAR domain across the 1,765 split PAAR sequences and constructed a phylogenetic tree using the same methods as done for analysis in Figure 1. We found that all orphan PAAR sequences grouped at the root of the tree suggesting that they are a common ancestor to the PAAR domain of prePAAR-containing effector sequences (added in new Figure 6—figure supplement 2 and updated Materials and methods).

f) In the Abstract, the authors state "…our findings define the mechanism of chaperone assisted secretion of a widespread family of T6SS membrane protein effectors". It appears that the significant part of this claim is already made in their earlier work (Quentin et al., 2018) and therefore the authors need to make a clear distinction.

We agree with the reviewer. To address this, we have changed the last sentence of the Abstract to the following: “Taken together, our findings reveal mechanisms of chaperone-mediated stabilization and secretion of two distinct families of T6SS membrane protein effectors.”

g) The authors have shown that mutations in pre-PAAR affect the binding of the effector to VgrG. However, there is no evidence which shows that prePAAR has any direct role in PAAR domain folding and therefore the lines in the Abstract, "…we find that prePAAR functions to facilitate proper folding of the downstream PAAR domain…" needs to be looked at.

We have changed the sentence to the following: “In addition to participating in the chaperone-TMD interface, we find that prePAAR residues mediate effector-VgrG spike interactions.”

h) In Abstract and Introduction, the author should also mention the number of genera which are spanned by these 6000 putative TMD containing effectors as this number does not directly signify the impact of the work.

We agree that the number of effectors is only significant if we state the number of genera these are found in. Therefore, we have changed the Abstract to include the line “encoded predominantly by 15 genera of Proteobacteria” to better highlight the widespread distribution of these effectors.